# Conflict-Averse IL-RL: Resolving Gradient Conflicts for Stable Imitation-to-Reinforcement Learning Transfer

## Abstract

Reinforcement Learning (RL) and Imitation Learning (IL) offer complementary capabilities: RL can learn high-performing policies but is data-intensive, whereas IL enables rapid learning from demonstrations but is limited by the demonstrator's quality. Combining them offers the potential for improved sample efficiency in learning high-performing policies, yet naïve integrations often suffer from two fundamental issues: (1) *negative transfer*, where optimizing the IL loss hinders effective RL fine-tuning, and (2) *gradient conflict*, where differences in the scale or direction of IL and RL gradients lead to unstable updates. We introduce *Conflict-Averse IL-RL* (CAIR), a general framework that addresses both challenges by combining two key components: (1) *Loss Manipulation*: an adaptive annealing mechanism utilizing a convex combination of IL and RL losses. This mechanism dynamically increases the weight of the RL loss when its gradient aligns with the IL gradient and decreases it otherwise, mitigating instabilities during the transition from IL to RL. (2) *Gradient Manipulation*: to further reduce conflict, we incorporate CAGrad to compute a joint gradient that balances IL and RL objectives while avoiding detrimental interference. We show that under trust-region assumptions (KL-proximity, and surrogate-improvement), CAIR guarantees monotonic non-deterioration in the expected return when the loss weights are annealed monotonically. To demonstrate its effectiveness, we evaluate CAIR on two benchmark suites: (1) sparse-reward robotic manipulation benchmarks (Fetch and Adroit), which require demonstration data for meaningful learning, and (2) dense-reward continuous control locomotion benchmark (D4RL), where demonstrations serve to accelerate standard RL. Compared against relevant hybrid IL–RL baselines, CAIR demonstrates competitive performance across both sparse- and dense-reward tasks. While CAIR does not achieve the strongest result on every individual task, its best reported variants attain the highest aggregate normalized reward among the evaluated methods under the task-specific reference scores used in this study. Evaluations across multiple IL (BC, DAgger) and RL (DDPG, SAC, PPO) combinations demonstrate that CAIR can be instantiated with diverse IL–RL pairings, while also showing that performance remains pairing- and domain-dependent, with no single combination consistently dominating. Among the evaluated pairings, BC+PPO is generally weaker than those using off-policy RL algorithms.

## 1 Introduction

Recent advancements in automation demonstrate a shift from traditional rule-based controllers to adaptive, AI-driven controllers in applications such as data center cooling (Heimerson et al., 2022), traffic management (Ault & Sharon, 2021; Ault et al., 2020), chatbots (Li et al., 2016), and self-driving cars (Kiran et al., 2021). These systems are often modeled as Markov Decision Processes (MDPs), enabling the use of Reinforcement Learning (RL) to optimize control policies. RL aims to learn the optimal policy but often requires substantial training data, which can be prohibitively expensive in many applications. By contrast, Imitation Learning (IL) offers a sample-efficient alternative where existing sub-optimal controllers or human demonstrations are leveraged to quickly train a competent controller. However, the competency of a controller trained via IL is often limited by the quality of the demonstrations used to train it.

In recent years, researchers have proposed combined IL and RL frameworks aiming to achieve the best of both worlds: sample efficiency and optimal asymptotic performance. Such approaches differ in how they incorporate demonstrator feedback—e.g., using demonstration-based rewards (Kang et al., 2018; Bajaj et al., 2023), replay experience relabeling (Nair et al., 2018; Zhu et al., 2022), or goal-based feedback (Nair et al., 2018)—and in the type of demonstrations utilized, such as static datasets or interactive feedback (Ziebart et al., 2008; Warnell et al., 2018; Christiano et al., 2017). Despite recent progress in integrating IL and RL, two significant challenges remain: (1) *Negative Transfer*—where naïve sequencing of IL followed by RL may force the RL policy to relearn or even discard imitation guidance, degrading the RL agent's ability to converge to the optimal policy compared to learning from scratch (Taylor & Stone, 2009; Zhang et al., 2023); and (2) *Gradient Conflict*—when IL and RL objectives are optimized simultaneously, their gradients may conflict in scale or direction (negative cosine similarity), leading to unstable or suboptimal updates. Gradient conflicts are particularly severe when IL relies on suboptimal demonstrations, as updates may push the policy away from the optimal RL solution. Such misalignment can result in learning instabilities and might contribute to negative transfer (Taylor & Stone, 2009).

Traditional annealing-based approaches mitigate negative transfer by considering a convex loss combination over the IL and RL objectives. They typically decay the weight of the IL component over fixed schedules (Rengarajan et al., 2022; Goecks et al., 2019; Zhu et al., 2022). However, such heuristics require careful tuning and might still produce adverse policy updates, especially when the IL loss gradients conflict with, or are overwhelmed by, the RL loss gradients. To address this limitation, we introduce *Conflict-Averse IL-RL (CAIR)*. CAIR introduces a principled *adaptive loss annealing* framework that optimizes an *alignment objective* rather than relying on predefined annealing schedules. Specifically, CAIR defines the weights of the convex loss combination via a constrained optimization formulation: it sets the weights such that the gradient of the combined loss (denoted the *combined gradient*) maximizes alignment with the RL gradient, subject to the constraint that the cosine similarity between the combined gradient and the IL gradient remains positive.

While the proposed adaptive loss annealing is designed to reduce harmful gradient conflict, it does not guarantee non-negative alignment at every update, for example when the constrained problem is temporarily infeasible or is solved only approximately. To further mitigate residual disagreement, we incorporate Conflict-Averse Gradient Descent (CAGrad) (Liu et al., 2021), which adjusts the joint update to improve its worst-case alignment with the IL and RL gradients. In the representative analyses presented in Section 3.3, combining adaptive annealing with CAGrad reduces negative alignment and improves the measured worst-case gradient alignment. These results support CAIR as a mechanism for mitigating negative transfer arising from gradient interference during the transition from imitation-driven to reward-driven optimization.

We present this framework in detail and show that it is theoretically grounded, admitting a conditional monotonic non-deterioration result for policy sequences satisfying the stated trust-region assumptions (KL-proximity of the policy sequence, and surrogate-improvement) (Schulman et al., 2017; 2015b). Empirical evaluations on sparse-reward MuJoCo environments (Todorov et al., 2012) demonstrate the effectiveness of CAIR across diverse RL algorithms, including off-policy methods such as SAC (Haarnoja et al., 2018) and DDPG (Lillicrap et al., 2015), as well as the on-policy method PPO (Schulman et al., 2017). The results show that CAIR can be effectively instantiated with diverse IL–RL pairings, while also demonstrating that the strongest-performing pairing varies across benchmark tasks i.e., no combination universally dominates the others. On the other hand, the on-policy BC+PPO combination is demonstrated to consistently underperform the other combinations. Relative to the evaluated hybrid IL–RL baselines, CAIR consistently achieves competitive performance on the sparse-reward manipulation tasks, attaining the strongest overall results in a majority of the evaluated tasks while learning policies that match or exceed demonstrator performance across all five tasks. To evaluate whether these benefits extend beyond sparse-reward settings, we further study six D4RL (Fu et al., 2020) locomotion tasks. The results demonstrate that CAIR remains competitive in dense-reward locomotion settings, achieving the strongest overall performance on multiple tasks while remaining comparable to established hybrid IL–RL baselines on others. Overall, the results show that two variants of CAIR (DAgger+DDPG and BC+SAC) provide competitive performance relative to hybrid IL–RL baselines across both benchmark suites.

## 2 Preliminaries

**Problem Formulation.** We consider control problems modeled as a Markov Decision Process (MDP) defined by the tuple $\langle \mathcal{S}, \mathcal{A}, \mathcal{P}, R, \gamma \rangle$, where $\mathcal{S}$ is the state space, $\mathcal{A}$ is the action space, $\mathcal{P} : \mathcal{S} \times \mathcal{A} \times \mathcal{S} \to [0, 1]$ is the transition function, $R : \mathcal{S} \times \mathcal{A} \to \mathbb{R}$ is the reward function, and $\gamma$ is the discount factor. At each timestep $t$, the agent executes an action $a_t$ in state $s_t$, leading to a new state $s_{t+1}$ and a reward $r_t = R(s_t, a_t)$. The tuple $(s_t, a_t, r_t, s_{t+1})$ is denoted as a *transition*. A finite sequence of successive transitions starting from an initial state $s_0$ and leading to a terminal state $s_T$ forms an *episode*, denoted by $\tau = (s_0, a_0, r_0, s_1, \ldots, s_{T-1}, a_{T-1}, r_{T-1}, s_T)$.

**Reinforcement Learning (RL).** Given an MDP, the Reinforcement Learning (RL) objective is to learn a policy $\pi$ that maximizes the expected return. A policy is defined either as a mapping from states to actions $\pi : \mathcal{S} \to \mathcal{A}$ (deterministic policy) or as a mapping from states to a probability simplex over the action space $\pi : \mathcal{S} \to \Delta(\mathcal{A})$ (stochastic policy). The expected return is defined as $J(\pi) = \mathbb{E}_{\tau \sim \pi} \left[ \sum_{t=0}^{\infty} \gamma^t R(s_t, a_t) \right]$. The optimal policy is defined by $\pi^* = \underset{\pi}{\arg\max} \; J(\pi)$. We consider the policy to be a function approximator (e.g., a neural network) with tunable parameters $\theta$, denoted by $\pi_\theta$. We evaluate both sparse-reward manipulation and dense-reward locomotion problems.

**Imitation Learning (IL).** Imitation Learning (IL) aims to learn control policies directly from a demonstrator, bypassing the need for a reward function. IL assumes an MDP $(\mathcal{S}, \mathcal{A}, \mathcal{P}, \gamma)$ *without* a reward function, together with either a fixed set of offline demonstrations or, in interactive IL (Ross et al., 2011), access to a demonstrator policy.

**Offline Demonstration.** A dataset $\mathcal{D} = \{\tau_i\}$ of trajectories generated by a demonstrator is available offline.

**Demonstrator Query.** An oracle $\pi^d$ is available for querying demonstrator actions given a state.

Note that querying a demonstrator does not necessitate that the demonstrator function is known, only that it can be queried. In this sense, a human can be considered an interactive demonstrator. Moreover, the underlying demonstrator policy may be stochastic; the assumption simply implies that an action sample is provided rather than a full action distribution. Finally, the demonstrator—in either the offline or online case—is not assumed to act optimally. Indeed, in this work, we focus on sub-optimal demonstrations that necessitate RL fine-tuning beyond the IL objective.

The objective in IL is to learn a policy that matches the demonstrator's behavior. IL is commonly formulated as minimizing a divergence between trajectories generated by the learner and those in the demonstration dataset:

$$L_{\mathrm{IL}}(\theta) = \mathbb{E}_{\tau_\pi \sim \pi, \, \tau_i \sim \mathcal{D}} \big[ \mathrm{Div}(\tau_\pi, \tau_i) \big],$$

where Div measures trajectory- or state-action-level mismatch. A widely used special case is *Behavioral Cloning* (BC) (Bain & Sammut, 1995; Torabi et al., 2018), which reduces IL to supervised learning. Here the divergence is defined at the one-step action level as:

$$\mathrm{Div}_{\mathrm{BC}}(\tau_\pi, \tau_i) = - \sum_{(s, a^d) \in \tau_i} \log \pi_\theta(a^d \mid s),$$

which corresponds to the negative log-likelihood of demonstrated actions under the learner policy. IL methods are typically more *sample efficient* than RL because they rely solely on demonstrations without requiring environment exploration. However, IL performance is strongly influenced by the quality and coverage of the demonstrations.

**Combined IL–RL Loss.** Access to both the reward function $R$ and demonstrations $\mathcal{D}$ (or an interactive demonstrator $\pi^d$) enables the integration of imitation and reinforcement learning objectives. IL can guide early learning in sparse-reward settings, while RL drives the policy toward maximizing expected return.

A weighted combination of IL and RL losses provides a common and flexible way to unify these objectives. Such formulations have been used with on-policy trust-region RL algorithms (Rengarajan et al., 2022; Kang et al., 2018), and off-policy actor-critic algorithms (Hester et al., 2018; Nair et al., 2020; Zhu et al., 2022). Because the weighted formulation does not require a particular functional form for $L_{\mathrm{IL}}$ or $L_{\mathrm{RL}}$, beyond

both yielding gradients with respect to the shared policy parameters, it provides a general mechanism for transitioning between the two objectives. Formally, we define the combined objective as:

$$L_\lambda(\theta) = (1 - \lambda) L_{\text{RL}}(\theta) + \lambda L_{\text{IL}}(\theta), \qquad \lambda \in [0, 1]. \tag{1}$$

**Gradient Conflict and Negative Transfer.** To analyze how the combined objective in equation 1 behaves during optimization, we examine the relationships between the gradients of the IL, RL, and combined losses, denoted by $g_{\text{IL}} = \lambda \nabla_\theta L_{\text{IL}}(\theta)$, $g_{\text{RL}} = (1 - \lambda) \nabla_\theta L_{\text{RL}}(\theta)$, and:

$$g_\lambda = \nabla_\theta L_\lambda(\theta) = g_{\text{RL}} + g_{\text{IL}}.$$

We measure *alignment* using the inner product or its normalized form (cosine similarity), where $\langle g_i, g_j \rangle > 0$ indicates aligned updates, and $\langle g_i, g_j \rangle < 0$ indicates conflicting updates.

*Gradient Conflict* arises when the IL and RL gradients disagree, i.e., $\langle g_{\text{IL}}, g_{\text{RL}} \rangle < 0$, indicating that the two objectives recommend incompatible parameter updates. In such cases, their convex combination $g_\lambda$ can also become negatively aligned with one of the objectives, leading to unstable or ineffective updates.

*Negative Transfer* is classically defined as a degradation in learning caused by using additional data or objectives, relative to learning without them (Taylor & Stone, 2009). In IL–RL training, this corresponds to a drop in performance when transitioning from IL-dominated to RL-dominated optimization as $\lambda$ decreases. However, such performance drops are typically difficult to predict *a priori* and are often only visible after many updates (Taylor & Stone, 2009; Taylor, 2009; Wang et al., 2021). Consequently, the literature lacks a general-purpose mechanism for detecting and avoiding negative transfer at the level of individual optimization updates, particularly when multiple objectives are jointly optimized.

In this work, we focus on gradient conflicts as a plausible cause for negative transfer. Specifically, in the IL–RL setting, we flag an update as *susceptible to negative transfer* when the combined gradient moves against the IL gradient, i.e., $\langle g_{\text{IL}}, g_\lambda \rangle < 0$. When this occurs, the resulting update direction may move the policy away from demonstrator-guided behavior. Since the demonstrations are precisely what provide a strong initialization and help overcome sparse-reward exploration difficulties, such anti-aligned updates may cause drops in performance.

This characterization is specific to the IL–RL combined-objective setting and is not intended as a universal definition of negative transfer. Rather, it targets one important mechanism by which negative transfer can arise—destructive interference between IL and RL update directions—and, crucially, provides a *pre-update* condition that we can enforce by manipulating $L_\lambda$ and its gradients. This is in line with findings from multi-task learning, where gradient cosine similarity is widely used to quantify task interference (Liu et al., 2021; Jiang et al., 2023), even though it does not explain all forms of negative transfer (Jiang et al., 2023).

## 2.1 Related Work

**Combining IL and RL.** Prior work has explored the integration of IL and RL objectives through linear combinations. For instance, *LOGO* (Rengarajan et al., 2022), *POfD* (Kang et al., 2018), and *Reward Phasing* (Bajaj et al., 2023) leverage demonstration data with trust-region-based guarantees but are often limited to on-policy gradient algorithms that use constrained optimization (Schulman et al., 2017; 2015b). Alternatively, off-policy training methods define implicit curricula through shifted sample distributions (Hester et al., 2018; Nair et al., 2018; Zhu et al., 2022), though they typically lack formal convergence guarantees and rely on heuristically tuned schedules. In contrast, we propose CAIR, a principled framework that supports both on-policy and off-policy RL algorithms combined with either offline or interactive IL methods. When instantiated with a trust-region solver, CAIR additionally admits a monotonic non-deterioration result (under trust-region assumptions Schulman et al. (2017; 2015b)) during a monotonic IL-to-RL transition.

**Offline and Hybrid RL.** Offline RL learns reward-driven policies from fixed datasets without additional environment interaction (Levine et al., 2020). Although some offline RL methods incorporate auxiliary imitation losses (Hester et al., 2018; Nair et al., 2018; 2020), these are typically used as regularizers rather than as part of an explicit IL–RL optimization framework. A common focus of offline RL is mitigating

distributional shift and out-of-distribution action selection, through conservative value-learning approaches such as CQL (Kumar et al., 2020) and IQL (Kostrikov et al., 2021). Hybrid RL methods combine offline datasets with online interaction to improve the sample efficiency of the RL agent (Song et al., 2022; Ball et al., 2023). Similar to offline RL, these methods tend to focus on stabilizing value learning under distribution shift. The offline and hybrid RL baselines considered here do not explicitly manipulate gradient conflict between imitation and reinforcement objectives—a challenge our method, CAIR, is designed to address directly.

**Negative Transfer.** Negative transfer broadly refers to situations in which incorporating additional data or objectives degrades learning performance relative to training without them (Taylor & Stone, 2009). In IL–RL settings, it commonly arises during the transition from imitation-dominated to reward-dominated optimization, where imitation-driven policy updates reduce the sample efficiency or asymptotic performance of the RL agent (Goecks et al., 2019; Rengarajan et al., 2022). Some prior approaches mitigate this effect using fixed or hand-designed schedules for annealing the IL update weight (Rengarajan et al., 2022; Goecks et al., 2019; Zhu et al., 2022). Such approaches are commonly agnostic to IL–RL gradient interactions and can still produce adverse update directions. Recent work addresses negative transfer by detecting when auxiliary signals harm the main objective: Du et al. (2018) adapt auxiliary-loss weights using gradient cosine similarity to reduce interference with a fixed primary task, though their method does not model a staged IL–to–RL transition and may downweight imitation losses prematurely in sparse-reward settings. Wang et al. (2019) mitigate negative transfer in transfer learning by filtering irrelevant source data. ForkMerge (Jiang et al., 2023) proposes a validation-driven branching strategy to manage task interference.

**Gradient Manipulation in Multi-Task Learning.** Conflicting gradients have been previously addressed in the context of Multi-Task Learning (MTL) through gradient manipulation methods (Désidéri, 2012; Yu et al., 2020; Sener & Koltun, 2018; Liu et al., 2021; 2023), which resolve gradient conflicts through joint objective optimization. Notably, CAGrad (Liu et al., 2021) provably converges to the optimum of the average loss, unlike MGDA (Sener & Koltun, 2018) and PCGrad (Yu et al., 2020), which may converge to arbitrary Pareto stationary points. Our proposed approach, CAIR, builds on this perspective by framing IL–RL integration as a dynamic MTL problem, justifying the use of CAGrad to resolve gradient conflicts between IL and RL objectives. This formalization enables mitigation of gradient conflicts through per-update adaptive objective weighting and gradient manipulation.

While we target gradient conflict as a primary contributing factor to negative transfer in IL–RL training, we acknowledge that other factors may also contribute (Jiang et al., 2023).

# 3   Conflict-Averse IL to RL

In this section, we formally present our framework, *Conflict-Averse IL-RL* (CAIR). CAIR addresses negative transfer when optimizing a convex combination of IL and RL by reducing harmful gradient interference during the transition from IL-dominated to RL-dominated optimization.

We achieve this through two complementary mechanisms: (1) **Adaptive Loss Annealing**, which dynamically reweights the IL and RL losses to maximize RL contributions only when they align with the IL objective; and (2) **Conflict-Averse Gradient Descent (CAGrad)**, which projects the update direction to balance both gradients even when their magnitudes differ. Together, these mechanisms enable the transition from IL to RL to avoid regressions caused by gradient misalignment.

## 3.1   Adaptive Loss Annealing

Algorithms that combine IL and RL typically prioritize the IL objective in early training to quickly imitate the demonstrator, before shifting focus to RL to optimize expected returns $J(\pi)$ (Kober et al., 2010; Rajeswaran et al., 2017). However, this sequential transition is prone to *negative transfer*, where the policy experiences a drop in performance during the shift from IL-dominated to RL-dominated training.

As shown in Figure 1(a), optimizing a convex combination of IL and RL losses illustrates tension between the two training objectives on the *FetchPush-v1* task (see Appendix A.1 for experimental details). We observe that reducing the IL loss often increases the RL loss and vice versa. This mismatch indicates that

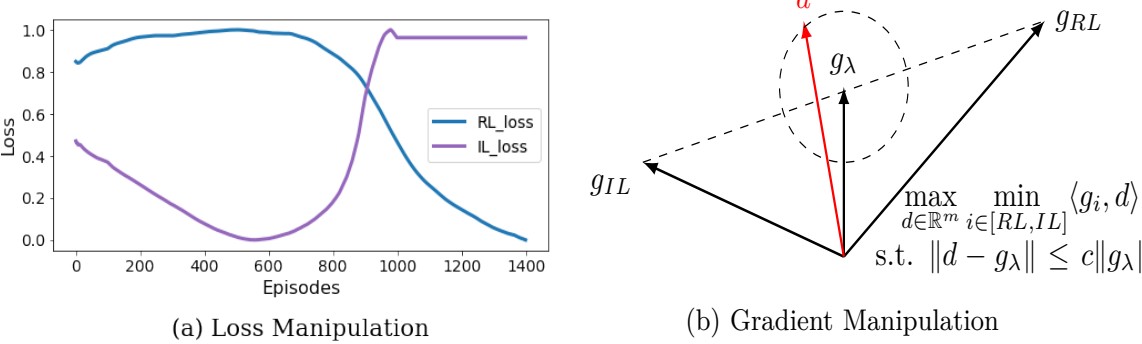

(a) Loss Manipulation        (b) Gradient Manipulation

Figure 1: **(a)** Evolution of Soft Actor-Critic (RL) and Behavioral Cloning (IL) losses on the *FetchPush-v1* task using standard linear annealing ($\lambda : 1 \rightarrow 0$) without gradient manipulation. As the IL weight decreases, the RL loss decreases while the IL loss spikes, suggesting the objectives are conflicting. (x-axis: Episodes $\times 10^3$; y-axis: Normalized loss). **(b)** The optimization landscape for the CAGrad method. $g_\lambda$ represents the naive combined gradient, while $d$ is the optimized update direction found by projecting $g_\lambda$ to maximize the minimum inner product between $g_{IL}$ and $g_{RL}$, while remaining within the allowed deviation from $g_\lambda$ (depicted by the acceptance region within the dashed circle).

naïve IL-to-RL transitions can drive updates away from imitation-induced behavior, leading to temporary performance degradation.

In contrast to prior work that anneals the IL weight $\lambda$ over fixed heuristics (Rengarajan et al., 2022; Goecks et al., 2019; Zhu et al., 2022), we propose a principled mechanism that *optimizes an alignment objective* to select $\lambda$ dynamically. Specifically, we choose $\lambda$ by solving:

$$\max_{\lambda \in [0,1]} \cos(\tilde{g}_\lambda, \tilde{g}_{\mathrm{RL}}) \quad \text{s.t.} \quad \cos(\tilde{g}_\lambda, \tilde{g}_{\mathrm{IL}}) \geq 0. \tag{2}$$

Here, $\tilde{g}_{\mathrm{IL}}$ and $\tilde{g}_{\mathrm{RL}}$ denote the $L_2$-normalized IL and RL gradients, and $\tilde{g}_\lambda$ is the normalized convex combination used for alignment checks. We employ $L_2$-normalization to ensure that $\lambda$ is optimized based solely on directional alignment, preventing bias from differing gradient magnitudes (e.g., if the RL loss scale is significantly larger than the IL loss scale).

The constraint in Equation 2 is designed to keep the combined direction non-negatively aligned with the IL gradient whenever a feasible solution is found, thereby limiting updates that oppose the demonstrator-guided direction. Simultaneously, the maximization objective favors progress toward the RL goal. This enables a stable transition, allowing $\lambda$ to decrease only when the IL and RL gradients are sufficiently aligned. However, if the gradients do not sufficiently align, the constraint prevents further annealing, maintaining a nonzero $\lambda$ and preserving the influence of the imitation objective.

By framing loss annealing as an optimization problem rather than a fixed schedule, CAIR dynamically balances IL and RL contributions based on the local geometry of the loss landscape. This is particularly important in settings where RL gradients may be noisy or weak, as it avoids prematurely discounting imitation signals that are still essential for effective learning. The $\lambda$ optimization procedure is summarized in Algorithm 2 (Appendix A.4).

The effect of $\lambda$-annealing can be understood geometrically through the gradient-alignment illustration in Figure 2. Panel (a) shows the case where IL and RL gradients are well aligned: the angle between $g_{\mathrm{IL}}$ and $g_{\mathrm{RL}}$ is small, and the combined gradients $g_{\lambda_{\mathrm{high}}}$ (solid orange) and $g_{\lambda_{\mathrm{low}}}$ (dashed brown) remain nearly identical. In this case, decreasing $\lambda$ produces consistent update directions, and the transition from IL to RL can proceed aggressively without destabilizing learning.

Conversely, Panel (b) represents the situation where gradients strongly conflict. Here, the direction of $g_\lambda$ becomes highly sensitive to the choice of $\lambda$: a small decay can rotate the update direction dramatically toward

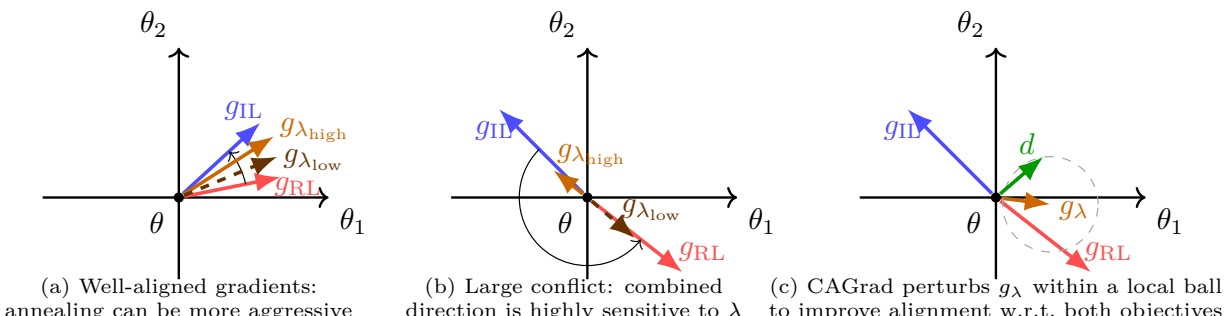

Figure 2: Geometric interpretation of loss annealing and gradient manipulation. Each panel shows a 2-D policy-parameter space $(\theta_1, \theta_2)$ with IL gradient $g_{\text{IL}}$ (blue), RL gradient $g_{\text{RL}}$ (red), and combined gradient $g_\lambda$. $g_{\lambda_{high}}$ (orange) uses a larger $\lambda$ than $g_{\lambda_{low}}$ (dashed-brown). **(a)** When gradients are aligned, annealing $\lambda$ (orange to dashed brown) maintains a consistent direction. **(b)** When gradients conflict, small changes in $\lambda$ cause drastic shifts in update direction, risking negative transfer. **(c)** CAGrad calculates a corrected direction $d$ (green) within a gradient-space deviation constraint around $g_\lambda$ to maximize joint agreement.

the RL gradient, causing abrupt policy shifts. This motivates our strategy of updating $\lambda$ only intermittently and annealing it conservatively whenever the gradients are misaligned.

## 3.2 Gradient Manipulation

While our adaptive loss annealing is designed to keep the combined gradient $g_\lambda$ positively aligned with the IL gradient whenever a feasible solution exists, it does not strictly eliminate conflicts between the underlying IL and RL components. When $g_{\text{IL}}$ and $g_{\text{RL}}$ oppose each other, the resulting $g_\lambda$ may effectively cancel out or be dominated by a single objective, preventing meaningful policy updates. This issue is often overlooked in previous approaches that jointly optimize IL and RL objectives via simple linear scalarization (Nair et al., 2018; Rajeswaran et al., 2017; Rengarajan et al., 2022; Zhu et al., 2022).

To address this, we adapt *Conflict-Averse Gradient Descent* (CAGrad) (Liu et al., 2021). Originally developed to resolve gradient conflicts in Multi-Task Learning (MTL), we repurpose CAGrad for the hybrid IL-to-RL setting to treat the IL and RL objectives as competing components within the combined loss $L_\lambda(\theta)$ (defined in Equation 1). CAGrad seeks an update direction $d$ that maximizes the worst-case improvement across both objectives, subject to a constraint that keeps $d$ close to the original combined gradient $g_\lambda$. Formally, the optimization is defined as:

$$\max_{d \in \mathbb{R}^m} \min_{i \in \{RL, IL\}} \langle g_i, d \rangle \quad \text{s.t.} \quad \|d - g_\lambda\| \leq c\|g_\lambda\|. \tag{3}$$

Here, $c \in [0, 1)$ is a hyper-parameter controlling the size of the gradient-space deviation constraint (or allowable deviation). The vector $d$ represents the optimal update direction within a local ball centered at $g_\lambda$ that maximizes the projection onto both $g_{\text{RL}}$ and $g_{\text{IL}}$.

CAGrad explicitly optimizes the minimum improvement across objectives at each step. It provably converges to a stationary point of the average loss while ensuring each update satisfies the locality constraint and corresponds to a Pareto-stationary direction (Liu et al., 2021). The complete CAIR framework, integrating both adaptive loss annealing and CAGrad, is summarized in Algorithm 1 (Appendix A.4).

The geometric intuition is illustrated in Figure 2(c). When $g_{\text{IL}}$ and $g_{\text{RL}}$ point in nearly opposite directions, the naïve combination $g_\lambda$ (orange arrow) may itself be poorly aligned with both objectives. CAGrad searches within a bounded region (dashed circle) around $g_\lambda$ to find a direction $d$ (green arrow) that maximizes the minimum agreement with the underlying gradients. The resulting update is balanced, avoiding the scenario where one objective dominates or cancels the other.

### 3.3 Empirical Gradient Analysis

A central challenge in IL-to-RL transfer lies in how the *combined gradient* $g_\lambda$ evolves as the weighting parameter $\lambda$ decreases. To understand this, it is useful to conceptually decompose the IL-to-RL transition into three stages: (1) an initial stage where the IL signal provides meaningful guidance but the RL signal is not yet informative, e.g., due to sparse rewards and insufficient exploration; (2) an intermediate stage where both signals are meaningful and, assuming the demonstrations are task-relevant, are mostly aligned; and (3) a final stage where the demonstrations become counterproductive due to their suboptimality, and the RL signal dominates. During stages (1) and (2), we are particularly interested in avoiding situations where $g_\lambda$ becomes *negatively aligned* with the imitation gradient $g_{\mathrm{IL}}$, i.e., $\langle g_{\mathrm{IL}}, g_\lambda \rangle < 0$, as this indicates that the combined update moves the policy *against* the direction favored by the demonstrator, potentially unlearning useful imitation-induced behavior.

During stage (3), however, enforcing alignment with $g_{\mathrm{IL}}$ is neither necessary nor desirable, as the RL objective should be free to surpass the demonstrator. Recall from Section 2 that $g_{\mathrm{IL}} = \lambda \nabla_\theta L_{\mathrm{IL}}(\theta)$, so as $\lambda \to 0$, $g_{\mathrm{IL}}$ approaches the zero vector. Since the cosine similarity constraint $\langle g_{\mathrm{IL}}, g_\lambda \rangle \geq 0$ is trivially satisfied for any $g_\lambda$ when $g_{\mathrm{IL}} = \mathbf{0}$, the alignment constraint becomes inactive precisely when it should — as the transition to pure RL optimization is complete. This means CAIR naturally transitions to standard RL optimization in stage (3) without requiring any explicit mechanism to disable the alignment constraint.

Following this intuition, we empirically investigate the impact of CAIR and its two components—adaptive loss annealing and gradient manipulation—on gradient alignment, focusing on $g_{\mathrm{IL}}$, $g_{\mathrm{RL}}$, the combined gradient $g_\lambda$, and the final update direction $d$.

Results are presented for a single run on the *FetchPick&Place-v1* task. The *FetchPickPlace-v1* task requires a robotic arm to grasp an object and move it to a target location. While demonstrations follow a particular trajectory distribution, multiple trajectories can achieve the same goal state. RL optimization may therefore favor alternative solutions, which can lead to disagreements between the imitation and RL gradients during stage (3) of the IL–to–RL transition.

The presented results demonstrate gradient alignment for *Behavior Cloning* (BC) and *DDPG* as the IL and RL algorithms, respectively. Representative results for the *BC+SAC* pairing on *FetchPush-v1* are additionally provided in Figure 16 in the appendix. Similar qualitative trends were observed for other IL–RL combinations listed in Section 3.5. To preserve the clarity of the temporal gradient evolution, these figures show a single representative seed, since averaging across seeds can smooth the transient gradient conflict events that the analysis is intended to illustrate. For clarity, each row in Figure 3 contains two plots for the same setting: the left column shows cosine similarity with the imitation gradient, $\cos(g_{\mathrm{IL}}, \cdot)$, and the right column shows cosine similarity with the RL gradient, $\cos(g_{\mathrm{RL}}, \cdot)$. This layout enables direct comparison of alignment with both the IL and RL objectives during annealing.

Figure 3(a) illustrates the conflict effect under *fixed-schedule loss annealing*. Let $\Delta_\lambda$ denote the annealing step used to update $\lambda \leftarrow \lambda - \Delta_\lambda$. The blue curve corresponds to a slower schedule ($\Delta_\lambda = 0.05$), while the purple curve corresponds to a faster schedule ($\Delta_\lambda = 0.2$). In the fast-annealing case, the cosine similarity between the IL gradient and the combined gradient drops below zero, indicating that the update direction actively opposes the imitation objective. While the slower schedule (blue) exhibits less abrupt fluctuations, it remains a heuristic approach that does not explicitly account for the alignment between $g_{\mathrm{IL}}$, $g_{\mathrm{RL}}$, and $g_\lambda$.

We next examine whether gradient manipulation alone (using CAGrad) is sufficient to improve alignment between the IL and combined gradients. Figure 3(b) compares fixed-step annealing ($\Delta_\lambda = 0.2$) with and without CAGrad using a moderate constraint parameter ($c = 0.75$). In this setting, CAGrad (purple) successfully adjusts the update direction so that it remains positively aligned with the IL gradient, mitigating gradient conflict despite aggressive annealing. However, CAGrad's effectiveness depends on the allowable deviation from $g_\lambda$. Figure 3(c) shows the same comparison with a tighter constraint ($c = 0.2$). Here, the adjusted direction cannot deviate sufficiently from the combined gradient to resolve the conflict, and negative alignment with $g_{\mathrm{IL}}$ re-emerges. This demonstrates that CAGrad is sensitive to hyperparameter tuning and is not robust to large conflicts induced by rapid or poorly timed changes in $\lambda$.

Finally, Figure 3(d) illustrates the full CAIR approach by comparing adaptive loss annealing without CAGrad (blue) to adaptive loss annealing combined with CAGrad (purple), both with $c = 0.2$. Adaptive annealing alone improves alignment relative to fixed schedules but can still produce negatively aligned updates due to discrete update intervals or temporarily infeasible alignment constraints. In contrast, CAIR—by jointly adapting $\lambda$ and refining the update direction with CAGrad—maintains positive alignment with both the IL and RL gradients throughout training. These results highlight the complementary roles of the two components: adaptive loss annealing shapes the objective to avoid inducing severe conflicts, while CAGrad resolves residual gradient disagreements at the update level. Together, they enable CAIR to effectively mitigate gradient-induced negative transfer during the IL-to-RL transition.

### 3.4 Conditional Monotonic Non-Deterioration

**Setting.** We consider a finite sequence of scalarized objectives parameterized by

$$1 = \lambda_0 > \lambda_1 > \cdots > \lambda_N = 0,$$

where each $\lambda_k$ defines our $\lambda$-combined loss $L_{\lambda_k}$ (Equation 1). The sequence begins at $\lambda_0 = 1$ (pure IL) and terminates at $\lambda_N = 0$ (pure RL), representing the full IL-to-RL transition.

For each $\lambda_k$, CAGrad is applied to optimize $L_{\lambda_k}(\theta)$. Under Assumption 1 below, CAGrad guarantees convergence to a first-order stationary point of $L_{\lambda_k}(\theta)$, satisfying $\min_{t \leq T} \mathbb{E}\|\nabla L_{\lambda_k}(\theta_t)\|^2 \to 0$ as $T \to \infty$, where $t$ indexes CAGrad gradient update steps (Theorem 3.2 in (Liu et al., 2021)). The resulting stationary policy corresponds to a Pareto-stationary solution of the IL–RL bi-objective. We denote the resulting policy $\pi_k^*$. Upon convergence to $\pi_k^*$, the annealing weight is updated to $\lambda_{k+1}$ and the optimization of $L_{\lambda_{k+1}}(\theta_{k+1})$ is initialized at the current policy $\pi_k^*$, ensuring continuity across successive stages.

The following assumptions ensure monotonic non-decreasing expected return between consecutive stationary policies, i.e., between $\pi_k^*$ and $\pi_{k+1}^*$ along the annealing schedule.

**Assumption 1** (CAGrad Regularity). *The objectives $L_{\mathrm{IL}}(\theta)$ and $L_{\mathrm{RL}}(\theta)$ are differentiable and $H$-smooth, i.e.,*

$$\|\nabla L_i(\theta) - \nabla L_i(\theta')\| \leq H\|\theta - \theta'\|, \qquad \forall \theta, \theta', \quad i \in \{\mathrm{IL}, \mathrm{RL}\},$$

*and $L_{\lambda_k}(\theta)$ is bounded below for every $\lambda_k$.*

**Assumption 2** (Stagewise KL Proximity). *For each consecutive pair $\pi_k^*$ and $\pi_{k+1}^*$, the expected state-conditional KL divergence is bounded:*

$$\mathbb{E}_{s \sim \rho_{\pi_k^*}} \big[ D_{\mathrm{KL}}\big(\pi_k^*(\cdot \mid s) \,\big\|\, \pi_{k+1}^*(\cdot \mid s)\big) \big] \ \leq \ \delta,$$

*for some $\delta > 0$.*

**Assumption 3** (Minimum Surrogate Improvement). *For each consecutive pair $\pi_k^*$ and $\pi_{k+1}^*$, the advantage-based surrogate improvement under $\pi_{k+1}^*$ is uniformly lower bounded:*

$$\mathbb{E}_{s \sim \rho_{\pi_k^*}} \Big[ \mathbb{E}_{a \sim \pi_{k+1}^*(\cdot|s)} A_{\pi_k^*}(s, a) \Big] \ \geq \ \eta,$$

*for some $\eta > 0$ that holds uniformly across all $k = 0, \ldots, N - 1$.*

**Lemma 1** (Stagewise Surrogate–KL Dominance). *Let*

$$C_k \ := \ \frac{2\gamma}{(1-\gamma)^2}\, \epsilon_k, \qquad \epsilon_k \ = \ \sup_{s,a} |A_{\pi_k^*}(s, a)|,$$

*denote the per-stage constant appearing in the TRPO performance lower bound (Theorem 1 in Schulman et al. (2015a)), and let $C^* = \max_k C_k$. Under Assumptions 2–3, if $\eta \geq C^*\delta$, then for every $k = 0, \ldots, N - 1$,*

$$\mathbb{E}_{s \sim \rho_{\pi_k^*}} \Big[ \mathbb{E}_{a \sim \pi_{k+1}^*(\cdot|s)} A_{\pi_k^*}(s, a) \Big] \ \geq \ C_k\, \mathbb{E}_{s \sim \rho_{\pi_k^*}} \big[ D_{\mathrm{KL}}\big(\pi_k^*(\cdot \mid s) \,\big\|\, \pi_{k+1}^*(\cdot \mid s)\big) \big].$$

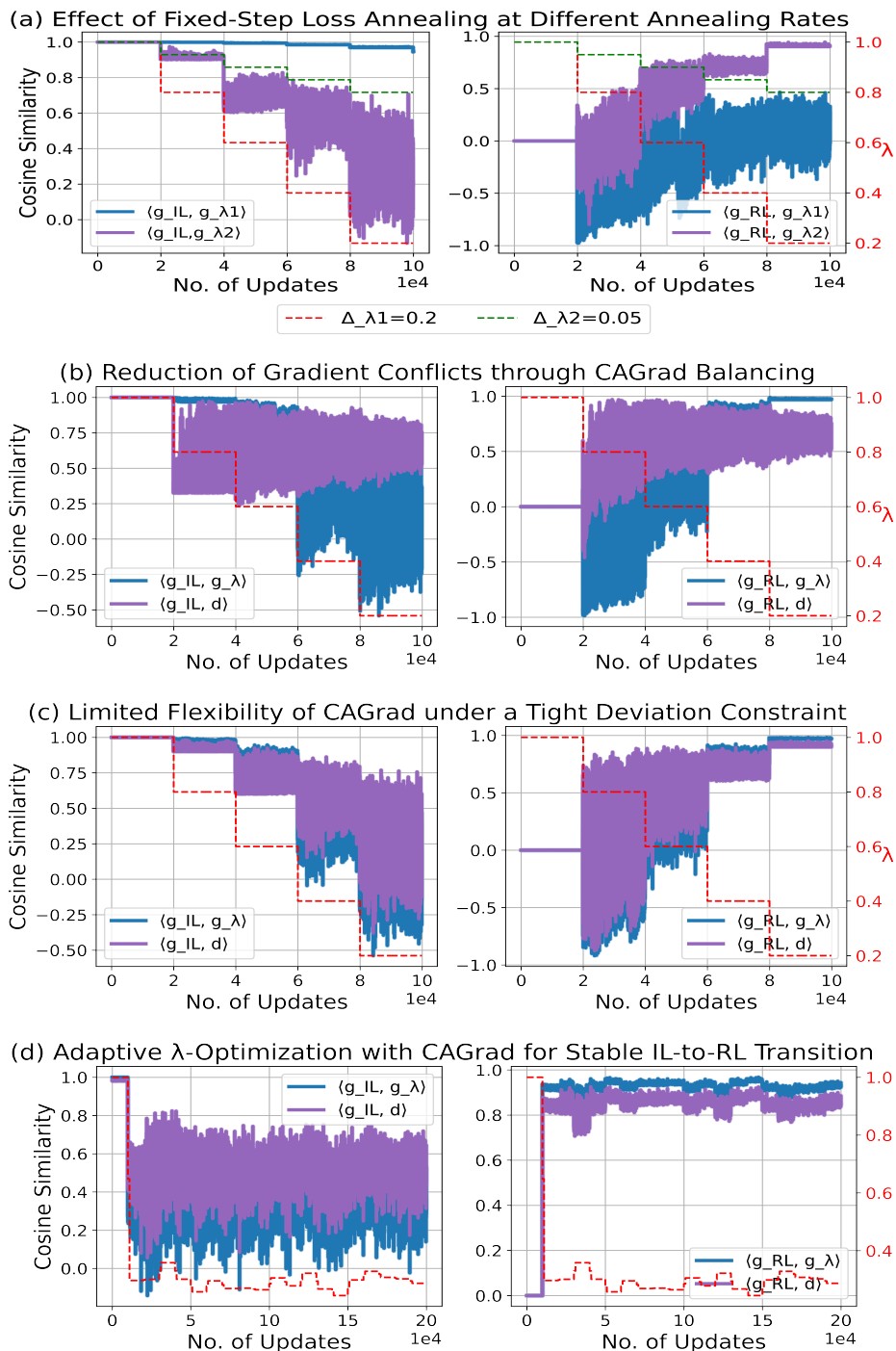

Figure 3: Cosine similarity analysis on *FetchPick&Place-v1* during the IL-to-RL transition. Each of the eight plots reports the alignment between the combined update direction and the task-specific (IL, RL) gradients: cosine similarity with $g_{\mathrm{IL}}$ (left column) and with $g_{\mathrm{RL}}$ (right column). **(a)** Naïve fixed annealing leads to negative alignment (conflict). **(b)** CAGrad with a loose constraint resolves conflict but requires tuning. **(c)** CAGrad with a tight constraint fails to resolve conflict under fixed annealing. **(d)** CAIR (Adaptive Annealing + CAGrad) maintains positive alignment, even under tight constraints. (x-axis: gradient updates; y-axis: cosine similarity averaged over 10 updates).

*Proof.* By Assumption 3, the left-hand side is at least $\eta$ for every $k$. By Assumption 2, the right-hand side satisfies

$$C_k \, \mathbb{E}_{s \sim \rho_{\pi_k^*}} \big[ D_{\mathrm{KL}}(\pi_k^* \| \pi_{k+1}^*) \big] \ \leq \ C_k \, \delta \ \leq \ C^* \delta \ \leq \ \eta,$$

which yields the claim. $\qquad\qquad\square$

**Theorem 1.** *Conditional Monotonic Non-Deterioration Under Assumptions 1–3 and the condition $\eta \geq C^* \delta$, the expected return is non-decreasing along the $\lambda$-schedule:*

$$J(\pi_{k+1}^*) \ \geq \ J(\pi_k^*), \qquad \forall\, k = 0, \ldots, N-1.$$

*Proof.* By Assumption 1 and Theorem 3.2 of (Liu et al., 2021), CAGrad converges to a stationary point of $L_{\lambda_k}(\theta)$, yielding a well-defined $\pi_k^*$. The TRPO performance lower bound (Theorem 1 in Schulman et al. (2015a)) gives

$$J(\pi_{k+1}^*) \ \geq \ J(\pi_k^*) + \mathbb{E}_{s \sim \rho_{\pi_k^*},\, a \sim \pi_{k+1}^*(\cdot|s)} \Big[ A_{\pi_k^*}(s,a) \Big] - C_k \, \mathbb{E}_{s \sim \rho_{\pi_k^*}} \big[ D_{\mathrm{KL}}\big(\pi_k^*(\cdot \mid s) \, \| \, \pi_{k+1}^*(\cdot \mid s)\big) \big].$$

By Lemma 1, the surrogate term dominates the KL penalty, so the right-hand side is at least $J(\pi_k^*)$, proving $J(\pi_{k+1}^*) \geq J(\pi_k^*)$. $\qquad\qquad\square$

**Discussion.** Assumption 2 (KL Proximity) is motivated by the CAIR optimization procedure, where optimization of $L_{\lambda_{k+1}}$ is initialized at $\pi_k^*$ (see Algorithm 1, Line 13). Since the two objectives $L_{\lambda_k}$ and $L_{\lambda_{k+1}}$ differ only in their scalarization weight, the resulting stationary point $\pi_{k+1}^*$ is expected to remain close to $\pi_k^*$, with the bound $\delta$ becoming tighter as the step size $\lambda_k - \lambda_{k+1}$ decreases. Assumption 3 (Minimum Surrogate Improvement) requires that the advantage of $\pi_{k+1}^*$ under the state distribution of $\pi_k^*$ is uniformly positive across all stages. This is reasonable since $\pi_{k+1}^*$ optimizes an objective with a larger RL weight than $\pi_k^*$, steering the policy locally toward higher expected return. The linking condition $\eta \geq C^* \delta$ ties Assumptions 2 and 3 together: it is satisfiable by choosing a sufficiently small $\delta$, i.e., a gradual enough $\lambda$-schedule, since $C^*$ is a fixed constant determined by the advantage function bounds. Finally, Theorem 1 ties these ingredients together: Assumption 1 guarantees that CAGrad converges to a well-defined stationary policy $\pi_k^*$ at each stage—without which Assumptions 2 and 3 would be vacuous—while the latter two assumptions, grounded in the TRPO performance bound, ensure that the resulting sequence of policies is non-deteriorating in expected return.

Theorem 1 should be interpreted as a conditional result for the stagewise policy sequence described above. It does not claim that CAIR itself enforces Assumptions 1–3; rather, it establishes that if the policy sequence produced during annealing satisfies these conditions, then its expected return is non-deteriorating. The result therefore formally applies only to policy updates satisfying the stated trust-region-style assumptions. Since SAC (Haarnoja et al., 2018) and DDPG (Lillicrap et al., 2015) do not formally enforce these conditions, the corresponding results in Section 4 provide empirical evaluation beyond the theorem's formal scope rather than an extension of its guarantee.

### 3.5 Generality of the Framework

Different pairings of IL and RL algorithms may behave differently in practice depending on the learning assumptions of the underlying methods. For example, off-policy RL algorithms such as SAC or DDPG may work well with offline IL methods such as Behavioral Cloning (Bain & Sammut, 1995; Torabi et al., 2018), whereas on-policy RL algorithms like PPO may align more naturally with interactive IL algorithms that collect data under the current policy, such as DAgger (Ross et al., 2011).

These considerations motivate examining multiple IL and RL combinations in our experiments. However, it is important to note that our primary goal is not to identify the single "best" pairing, but rather to demonstrate that CAIR can be instantiated with multiple IL–RL pairings. Its empirical performance remains dependent on the selected algorithms and task.

## 4  Experimental Study

Our experiments are designed to evaluate the performance of CAIR when paired with various combinations of IL and RL solvers. Specifically, we aim to answer the following research questions:

1. **Applicability:** Is CAIR applicable to different combinations of IL and RL algorithms? and **Generality:** Does CAIR provide consistent performance benefits across different combinations of IL and RL algorithms?

2. **Performance:** Can CAIR achieve improved sample efficiency and asymptotic performance compared to baseline hybrid-RL approaches?

3. **Ablation-Impact of CAGrad:** What is the relative impact of incorporating Conflict-Averse Gradient Descent (CAGrad) alongside adaptive loss annealing?

4. **Ablation-Impact of Adaptive Annealing:** What is the relative impact of incorporating adaptive loss annealing instead of fixed-schedule loss annealing?

### 4.1  Settings

**Benchmark Tasks.**   We conduct experiments across two broad classes of MuJoCo benchmark tasks (Todorov et al., 2012), categorized by their reward structure: sparse reward and dense reward.

**Sparse-Reward tasks.** This class presents significant exploration challenges due to sparse reward feedback and includes two sets of tasks:

- **Fetch Tasks** (Plappert et al., 2018): This set of tasks evaluates robotic arm manipulation. We use three tasks: *FetchPickAndPlace-v1* (P&P), *FetchSlide-v1* (FS), and *FetchPush-v1* (FP). These tasks require the agent to manipulate an object toward a desired goal location.

- **Adroit Tasks** (Rajeswaran et al., 2017): This set of tasks evaluates dexterous manipulation involving high-dimensional control of a simulated anthropomorphic hand. We use two tasks: *AdroitHandHammerSparse-v1* and *AdroitHandPenSparse-v1*, which feature highly complex dynamics.

**Dense-Reward Tasks.** Unlike the sparse manipulation tasks, this class uses dense reward functions to evaluate continuous control under high-dimensional dynamics. While not required in order to perform meaningful learning, demonstrations can facilitate better sample efficiency in such tasks.

- **D4RL Locomotion Tasks**: We evaluate continuous locomotion control using six tasks from the locomotion benchmark: *HalfCheetah*, *Hopper*, *Walker2d*, *Ant*, *Swimmer*, and *Humanoid*.

Episode lengths are capped at 1,000 steps for tasks in the Fetch and D4RL benchmark suites, and 4,000 steps for tasks in the Adroit benchmark suite.

**Demonstrations.**   Each task is associated with an offline dataset $\mathcal{D}$ of demonstration trajectories and an interactive demonstrator policy $\pi^d$ available for querying during training.

*Fetch tasks.* We use the suboptimal rule-based demonstrator introduced by Bajaj et al. (2023). This heuristic controller selects actions based on the relative Cartesian distances between the robot gripper, the manipulated object, and the target position. Although competent, the resulting $\pi^d$ does not produce optimal trajectories. The offline dataset $\mathcal{D}$ is constructed by executing $\pi^d$ in the environment and recording 50 resulting trajectories.

*Adroit tasks.* For each manipulation task, the demonstrations and the corresponding interactive demonstrator policy $\pi^d$ are obtained from the benchmark introduced by Rajeswaran et al. (2017). We use 50 demonstration trajectories per task.

*Locomotion tasks.* For the D4RL locomotion tasks, we use the `simple` Minari datasets (Younis et al., 2024), which provide offline demonstration trajectories generated by trained RL policies. Following the dataset generation procedure described in the Minari documentation, we reproduce the corresponding demonstrator policies and use them as the interactive demonstrator $\pi^d$ during training.

**Evaluation Protocol.** The main manipulation and locomotion evaluations use five and ten random seeds, respectively; where the number of available runs differs in an ablation, the exact sample size is reported in the corresponding table. We report the mean reward across runs with a 1-$\sigma$ shaded error region. For a horizon $H$, AUC@$H$ denotes the mean reward over the corresponding learning curve, AUC@$H = \frac{1}{H}\sum_{t=1}^{H} \bar{R}_t$, where $\bar{R}_t$ is the mean reward across seeds at evaluation point $t$. AUC is computed from the unsmoothed reward curve; smoothing is used only for visualization.

We additionally report aggregate normalized reward, NormReward $= \frac{R - R_{\min}}{R_{\text{demo}} - R_{\min}}$, using the task-specific reference scores reported in Table 11. For the manipulation tasks, $R_{\min}$ is a task-specific lower reward reference determined from the task reward scale. For the locomotion tasks, $R_{\min}$ is the lowest score observed among the evaluated methods and is therefore dependent on the evaluated method set. Values near 1 indicate performance similar to the demonstrator. We use normalized reward as a complementary cross-task summary and retain raw return as the primary task-level measure.

**Implementation.** We implement CAIR by extending the `stable-baselines3` library (Raffin et al., 2021). To support reproducibility, our codebase and execution instructions are publicly available.[1] Detailed hyperparameters and hardware specifications are provided in Appendix A.2.

## 4.2 Generality of CAIR

Our first set of experiments addresses Research Question 1: *Is CAIR applicable to different combinations of IL and RL algorithms and does it provide consistent performance improvement?*

To evaluate the applicability of CAIR across different IL–RL pairings, we instantiate the framework using multiple combinations of Imitation Learning (IL) and Reinforcement Learning (RL) algorithms. We pair two common IL approaches: (1) **Behavioral Cloning (BC)** (Bain & Sammut, 1995; Torabi et al., 2018), which learns from offline demonstrations; and (2) **DAgger** (Ross et al., 2011), which learns from an interactive demonstrator; with three distinct RL algorithms covering different learning paradigms: (1) **SAC** (Haarnoja et al., 2018) (stochastic off-policy); (2) **DDPG** (Lillicrap et al., 2015) (deterministic off-policy); and (3) **PPO** (Schulman et al., 2017) (stochastic on-policy).

**Implementation Details.** For both BC and DAgger, we use the Mean Squared Error (MSE) loss to minimize the divergence between the learner's action $a \sim \pi_\theta(\cdot|s)$ and the demonstrator's action $a_D = \pi_D(s)$:

$$L_{IL} = \frac{1}{N}\sum_{i=1}^{N} \|\pi_D(s_i) - a_i\|^2.$$

The annealing weight $\lambda$ is updated every $T_\lambda = 10$ episodes in the D4RL robotic manipulation tasks and every $T_\lambda = 5$ episodes in the locomotion tasks. [2] (see Line 22 of Algorithm 1 in Appendix A.4). Annealing is considered complete when $\lambda$ falls below a small threshold $\epsilon_\lambda < 0.02$; rather than checking for exact convergence to zero, we use a small positive threshold to account for approximation noise (see Line 11 of Algorithm 2 in Appendix A.4). Once this condition is met, $\lambda$ is fixed at zero and remains unchanged for the rest of training, ensuring that demonstrator guidance is fully withdrawn and the agent continues learning solely from environmental feedback.

**Results.**

**Manipulation Tasks.** Table 1 summarizes the final performance and area under the learning curve (AUC) achieved by different combinations of IL algorithms (BC, DAgger) and RL algorithms (SAC, DDPG, PPO) on the sparse-reward manipulation tasks. The corresponding learning curves are provided in Figure 5 in Appendix A.4. Across the evaluated pairings, all IL–RL combinations exceed demonstrator performance in 3/5 tasks. Among the evaluated pairings, BC+SAC achieves the strongest overall performance on *FetchSlide-v1*, *AdroitHandHammerSparse-v1*, and *AdroitHandPenSparse-v1*, while DAgger+SAC achieves the strongest

---

[1] https://osf.io/hukge/?view_only=8f303806d77e4faf83371c36f8c5b4b2
[2] While we found $T_\lambda = 10$ and $T_\lambda = 5$ to work well in the reported tasks, we acknowledge that this value might require tuning in other tasks.

Table 1: Learning performance of CAIR instantiated with different combinations of IL algorithms (BC, DAgger) and RL algorithms (SAC, DDPG, PPO) on sparse-reward manipulation tasks. Values report final reward (mean ± standard deviation), area under the curve (AUC), and normalized reward (NormReward), computed as $(R - R_{min})/(R_{demo} - R_{min})$, where $R_{min}$ is the minimum score observed across all evaluated methods; values above 1 indicate performance exceeding the demonstrator. Training budgets are shown beneath each task. Methods marked with † are significantly worse than the highest-mean method in the same task (Welch's t-test, $p < 0.05$). Among the remaining methods, AUC determines the highest-ranked method, which is bolded. The final rows report average NormReward across tasks.

| Task | Pairing | Mean | ± SD | AUC | NormReward |
|------|---------|------|------|-----|------------|
| *FetchPickPlace* (10,000k steps) | BC+DDPG | -13.18 | ± 4.26 | -18.03 | 2.45 |
| | BC+SAC | -19.52 | ± 4.88 | -29.78 | 2.03 |
| | BC+PPO | -12.61 | ± 1.90 | -20.20 | 2.49 |
| | DAgger+DDPG | -12.48 | ± 3.33 | -17.26 | 2.50 |
| | **DAgger+SAC** | **-12.88** | **± 4.75** | **-14.13** | **2.47** |
| | DAgger+PPO | -14.03 | ± 3.00 | -16.44 | 2.40 |
| *FetchPush* (10,000k steps) | BC+DDPG | -14.35 | ± 3.49 | -19.23 | 1.82 |
| | BC+SAC | -10.92 | ± 2.55 | -22.41 | 1.98 |
| | BC+PPO | -26.95 | ± 6.02 | -28.74 | 1.04 |
| | DAgger+DDPG | -12.05 | ± 2.18 | -17.18 | 1.90 |
| | **DAgger+SAC** | **-9.80** | **± 1.23** | **-15.29** | **2.02** |
| | DAgger+PPO | †-19.10 | ± 2.98 | -22.02 | 1.40 |
| *FetchSlide* (10,000k steps) | BC+DDPG | -38.74 | ± 3.38 | -38.78 | 2.32 |
| | **BC+SAC** | **-35.55** | **± 4.83** | **-37.15** | **3.05** |
| | BC+PPO | †-43.45 | ± 2.43 | -44.03 | 1.17 |
| | DAgger+DDPG | -37.84 | ± 3.69 | -40.38 | 2.30 |
| | DAgger+SAC | -36.77 | ± 2.72 | -41.64 | 2.37 |
| | DAgger+PPO | -38.24 | ± 2.35 | -40.97 | 2.22 |
| *AdroitHammer* (3,000k steps) | BC+DDPG | 1,354 | ± 287 | 718 | 29.87 |
| | **BC+SAC** | **1,426** | **± 356** | **1,024** | **31.43** |
| | BC+PPO | †6 | ± 0 | -1 | 0.58 |
| | DAgger+DDPG | 609 | ± 666 | 277 | 13.66 |
| | DAgger+SAC | 650 | ± 751 | 338 | 14.57 |
| | DAgger+PPO | 809 | ± 412 | 471 | 18.02 |
| *AdroitPen* (8,000k steps) | BC+DDPG | †2,593 | ± 411 | 1,500 | 0.74 |
| | **BC+SAC** | **7,041** | **± 744** | **6,419** | **2.01** |
| | BC+PPO | †48 | ± 68 | 193 | 0.01 |
| | DAgger+DDPG | †3,009 | ± 353 | 2,349 | 0.86 |
| | DAgger+SAC | †3,306 | ± 389 | 2,063 | 0.94 |
| | DAgger+PPO | †3,808 | ± 188 | 3,794 | 1.09 |
| *Average Normalized Reward Across Tasks* | BC+DDPG | | | | 7.44 |
| | **BC+SAC** | | | | **8.10** |
| | BC+PPO | | | | 1.06 |
| | DAgger+DDPG | | | | 4.25 |
| | DAgger+SAC | | | | 4.48 |
| | DAgger+PPO | | | | 5.03 |

overall performance on *FetchPickAndPlace-v1* and *FetchPush-v1*. Consistent with these task-level results, BC+SAC achieves the highest average normalized reward across tasks (8.10). BC+PPO is the weakest pairing, failing to match demonstrator performance on *AdroitHandHammerSparse-v1* and *AdroitHandPenSparse-v1* (Normalized Reward < 1.0), while BC+DDPG and DAgger+DDPG each fall short of the demonstrator in 1/5 tasks. Overall, SAC-based variants achieve the strongest performance across the manipulation tasks.

**Locomotion Tasks.** Table 2 summarizes similar measurements on the D4RL locomotion tasks. The corresponding learning curves are provided in Figure 6 in Appendix A.4. Across all six tasks, at least one IL–RL pairing matches or exceeds demonstrator performance (Normalized Reward $>\simeq 1.0$). However, no single IL–RL combination consistently achieves this across all six tasks. DAgger+DDPG achieves the highest-ranked performance on *HalfCheetah-v5*, *Hopper-v5*, *Swimmer-v5*, and *Humanoid-v5*, while DAgger+SAC achieves

Table 2: Learning performance of CAIR instantiated with different combinations of IL algorithms (BC, DAgger) and RL algorithms (SAC, DDPG, PPO) on D4RL locomotion tasks. Values report final reward (mean ± standard deviation), area under the curve (AUC), and normalized reward (NormReward), computed as $(R - R_{\min})/(R_{\text{demo}} - R_{\min})$, where $R_{\min}$ is the minimum score observed across all evaluated methods; values above 1 indicate performance exceeding the demonstrator. Methods marked with † are significantly worse than the highest-mean method in the same task (Welch's t-test, $p < 0.05$). Among the remaining methods, AUC determines the highest-ranked method, which is bolded. The final rows report average NormReward across tasks.

| Task | Pairing | Mean | ± SD | AUC | NormReward |
|---|---|---|---|---|---|
| *HalfCheetah* (250k steps) | BC+DDPG | †8,462 | ± 661 | 4,909 | 0.85 |
| | BC+SAC | †7,113 | ± 332 | 3,677 | 0.73 |
| | BC+PPO | †1,197 | ± 863 | 626 | 0.17 |
| | **DAgger+DDPG** | **9,584** | **± 1,119** | **6,649** | **0.96** |
| | DAgger+SAC | †8,504 | ± 348 | 3,929 | 0.86 |
| | DAgger+PPO | †2,963 | ± 617 | 1,890 | 0.33 |
| *Hopper* (250k steps) | BC+DDPG | †3,140 | ± 256 | 2,682 | 0.91 |
| | BC+SAC | 3,424 | ± 119 | 2,864 | 1.03 |
| | BC+PPO | †1,883 | ± 562 | 1,951 | 0.38 |
| | **DAgger+DDPG** | **3,328** | **± 210** | **2,944** | **0.99** |
| | DAgger+SAC | 3,363 | ± 49 | 2,876 | 1.01 |
| | DAgger+PPO | †3,102 | ± 306 | 3,021 | 0.90 |
| *Walker2d* (250k steps) | BC+DDPG | †1,200 | ± 743 | 1,038 | 0.32 |
| | BC+SAC | 3,814 | ± 325 | 2,534 | 0.93 |
| | BC+PPO | †2,161 | ± 489 | 1,870 | 0.55 |
| | DAgger+DDPG | 3,369 | ± 1,044 | 3,072 | 0.83 |
| | **DAgger+SAC** | **4,171** | **± 971** | **3,278** | 1.02 |
| | DAgger+PPO | †1,550 | ± 875 | 1,613 | 0.41 |
| *Swimmer* (250k steps) | BC+DDPG | †15.6 | ± 17.2 | 58.3 | 0.05 |
| | BC+SAC | †256.8 | ± 22.2 | 153.6 | 0.78 |
| | BC+PPO | †138.6 | ± 54.6 | 92.6 | 0.42 |
| | **DAgger+DDPG** | **306.1** | **± 68.5** | **289.3** | **0.93** |
| | DAgger+SAC | 318.0 | ± 12.6 | 284.8 | 0.96 |
| | DAgger+PPO | 331.5 | ± 4.4 | 275.1 | 1.00 |
| *Ant* (250k steps) | BC+DDPG | †1,085 | ± 1,402 | 97 | 0.64 |
| | BC+SAC | 3,013 | ± 1,252 | 1,465 | 1.04 |
| | BC+PPO | †1,270 | ± 442 | 1,233 | 0.66 |
| | DAgger+DDPG | †1,202 | ± 668 | 1,634 | 0.67 |
| | DAgger+SAC | 2,275 | ± 1,066 | 1,881 | 0.87 |
| | **DAgger+PPO** | **1,976** | **± 863** | **1,920** | **0.69** |
| *Humanoid* (250k steps) | BC+DDPG | †4,917 | ± 219 | 4,810 | 0.69 |
| | BC+SAC | 5,024 | ± 171 | 4,934 | 0.79 |
| | BC+PPO | †4,803 | ± 81 | 4,685 | 0.58 |
| | **DAgger+DDPG** | **5,346** | **± 97** | **5,248** | **1.09** |
| | DAgger+SAC | 5,416 | ± 266 | 5,026 | 1.15 |
| | DAgger+PPO | †4,957 | ± 84 | 4,881 | 0.73 |
| *Average Normalized Reward Across Tasks* | BC+DDPG | | | | 0.58 |
| | BC+SAC | | | | 0.88 |
| | BC+PPO | | | | 0.46 |
| | DAgger+DDPG | | | | 0.91 |
| | **DAgger+SAC** | | | | **0.98** |
| | DAgger+PPO | | | | 0.68 |

the highest-ranked performance on *Walker2d-v5*. DAgger+PPO achieves the highest-ranked performance on *Ant-v5*. Consistent with these task-level results, DAgger+SAC achieves the highest average normalized

reward across tasks (0.98), followed by DAgger+DDPG (0.91) and BC+SAC (0.88). Overall, DAgger-based variants achieve the strongest performance across the locomotion tasks.

Taken together, the manipulation and locomotion results provide a qualified positive answer to Research Question 1. CAIR can be instantiated with multiple IL–RL pairings across both sparse-reward manipulation and dense-reward locomotion tasks. However, performance differences between pairings remain significant, no single IL–RL pairing consistently achieves the strongest performance across all benchmark tasks. Across both benchmark suites, SAC-based variants achieve the strongest performance in the largest number of tasks. More specifically, BC-based variants achieve the strongest performance in the majority of the manipulation tasks, while DAgger-based variants achieve the strongest performance in the majority of the locomotion tasks.

**Sensitivity Analysis.** The previous experiments evaluate CAIR across different IL–RL pairings. We now examine the sensitivity of the proposed adaptive annealing and conflict-aware optimization mechanisms to hyperparameter selection. CAIR exposes three primary hyperparameters: (i) the CAGrad deviation parameter $c$; (ii) the update interval for the adaptive $\lambda$-optimization; and (iii) the learning rate $\eta_\lambda$ used for updating $\lambda$.

- **CAGrad Constraint ($c$):** Across tasks, we observe that moderate values (e.g., $c \in [0.45, 0.95]$) produce consistent behavior with only minor variations in final performance. However, when $c$ is set too small, the adjusted gradient is forced to remain close to the original combined gradient $g_\lambda$, preventing CAGrad from effectively resolving IL–RL conflicts. As observed in our ablation studies (Figure 9 in Appendix A.3), this can lead to severe performance degradation due to unmitigated gradient interference.

- **Update Interval:** To provide a stable learning signal, we update $\lambda$ every 10 episodes for the manipulation tasks and every 5 episodes for the locomotion tasks. This allows the policy to train under a fixed weight before recomputing the alignment balance. Updating too frequently makes the weighting overly sensitive to short-term gradient noise, whereas updating too infrequently produces a sluggish transition toward RL. The impact of this interval is detailed in Figures 12 and 13 (Appendix A.3).

- **Annealing Rate ($\eta_\lambda$):** The learning rate of the $\lambda$-optimizer exhibits a similar trade-off. Very small values stall the annealing process, while larger values produce abrupt shifts driven by noisy gradient estimates. Consequently, we default to $\eta_\lambda = 10^{-4}$. Figure 11 compares learning curves for $\eta_\lambda \in \{10^{-3}, 10^{-4}, 10^{-5}\}$, showing that both extremes reduce sample efficiency. This aligns with prior findings (Uchendu et al., 2022) that excessively rapid transitions from IL to RL can cause performance collapse, further motivating the need for controlled, alignment-aware annealing.

While several IL–RL pairings achieve demonstrator-level performance in the locomotion tasks, reaching demonstrator performance is only one aspect of successfully combining imitation learning and reinforcement learning. A policy trained through imitation can reproduce demonstrator behavior, but hybrid IL–RL methods must additionally maintain performance while gradually shifting optimization from the imitation objective to the RL objective. As shown in the adaptive annealing ablation studies (Section 4.6), this transition is non-trivial: policies can attain demonstrator-level performance and subsequently experience a decline in return as the influence of the imitation objective is reduced.

**Note on On-Policy Stability.** For on-policy methods like PPO, the transition is notably more brittle. We find that reducing the policy learning rate and entropy coefficient as $\lambda \to 0$ leads to more stable post-annealing performance. Furthermore, normalizing IL and RL gradients and maintaining a weighted moving average of their cosine similarity helps counteract high-variance gradient estimates, supporting a smoother transition into RL-dominated training.

### 4.3 Benchmark Comparison

Our next set of experiments addresses Research Question 2: *Can CAIR achieve improved sample efficiency and asymptotic performance compared to baseline hybrid-RL approaches?*

Table 3: Learning performance of CAIR (BC+DDPG and BC+SAC) compared to hybrid IL–RL baselines on sparse-reward manipulation tasks. Values report final reward (mean $\pm$ standard deviation), AUC, and normalized reward (NormReward), computed as $(R - R_{\min})/(R_{\mathrm{demo}} - R_{\min})$, where $R_{\min}$ is the minimum score observed across all evaluated methods; values above 1 indicate performance exceeding the demonstrator. Training budgets are shown beneath each task. Methods marked with † are significantly worse than the highest-mean method in the same task (Welch's t-test, $p < 0.05$). Among the remaining methods, AUC determines the highest-ranked method, which is bolded. The final rows report average NormReward across tasks.

| Task | Algorithm | Mean | $\pm$ SD | AUC | NormReward |
|---|---|---|---|---|---|
| *FetchPickPlace* (15,000k steps) | **BC+DDPG** | **-14.07** | **$\pm$ 5.50** | **-16.52** | **2.40** |
| | BC+SAC | -14.88 | $\pm$ 3.88 | -26.04 | 2.34 |
| | RP | -12.75 | $\pm$ 1.63 | -32.91 | 2.48 |
| | HER | †-27.14 | $\pm$ 2.63 | -26.80 | 1.52 |
| | SAIL | †-45.14 | $\pm$ 0.13 | -46.09 | 0.32 |
| | JumpStart | -14.40 | $\pm$ 3.18 | -27.68 | 2.37 |
| | RLPD | †-48.16 | $\pm$ 2.51 | -48.00 | 0.12 |
| *FetchPush* (10,000k steps) | **BC+DDPG** | **-13.57** | **$\pm$ 2.2** | **-18.12** | **1.82** |
| | BC+SAC | -10.39 | $\pm$ 1.8 | -20.64 | 1.98 |
| | RP | †-46.90 | $\pm$ 2.0 | -46.85 | 0.15 |
| | HER | †-22.62 | $\pm$ 2.1 | -23.79 | 1.37 |
| | SAIL | †-46.21 | $\pm$ 0.9 | -46.20 | 0.19 |
| | JumpStart | -23.53 | $\pm$ 14.2 | -28.99 | 1.32 |
| | RLPD | -12.78 | $\pm$ 5.1 | -20.20 | 1.86 |
| *FetchSlide* (18,500k steps) | BC+DDPG | †-38.84 | $\pm$ 2.86 | -38.75 | 2.23 |
| | **BC+SAC** | **-33.02** | **$\pm$ 3.34** | **-36.29** | **3.40** |
| | RP | †-39.10 | $\pm$ 2.12 | -44.34 | 2.18 |
| | HER | †-41.70 | $\pm$ 2.38 | -42.43 | 1.66 |
| | SAIL | †-47.44 | $\pm$ 0.03 | -47.72 | 0.51 |
| | JumpStart | †-44.65 | $\pm$ 1.80 | -45.32 | 1.07 |
| | RLPD | †-42.80 | $\pm$ 9.79 | -43.04 | 0.91 |
| *AdroitHammer* (3,000k steps) | BC+DDPG | 1,354 | $\pm$ 287 | 718 | 29.87 |
| | BC+SAC | 1,426 | $\pm$ 356 | 1,024 | 31.43 |
| | RP | †-19 | $\pm$ 1 | -19 | 0.00 |
| | HER+Demos | †14 | $\pm$ 37 | 13 | 0.75 |
| | SAIL | †10 | $\pm$ 30 | 12 | 0.65 |
| | JumpStart | †-15 | $\pm$ 7 | -12 | 0.05 |
| | **RLPD** | **1,358** | **$\pm$ 596** | **1,212** | **29.96** |
| *Pen* (8,000k steps) | BC+DDPG | †2,593 | $\pm$ 411 | 1,500 | 0.74 |
| | BC+SAC | 7,041 | $\pm$ 744 | 6,419 | 2.01 |
| | RP | †527 | $\pm$ 0 | 285 | 0.15 |
| | HER+Demos | †177 | $\pm$ 108 | 133 | 0.05 |
| | SAIL | †37 | $\pm$ 56 | 69 | 0.01 |
| | JumpStart | †3,636 | $\pm$ 2,104 | 4,372 | 1.04 |
| | **RLPD** | **7,565** | **$\pm$ 474** | **6,654** | **2.16** |
| *Average Normalized Reward Across Tasks* | DAgger+DDPG | | | | 7.41 |
| | **BC+SAC** | | | | **8.23** |
| | RP | | | | 0.99 |
| | HER+Demos | | | | 1.07 |
| | SAIL | | | | 0.34 |
| | JumpStart | | | | 1.17 |
| | RLPD | | | | 7.00 |

To answer this, we compare CAIR against the following common baseline algorithms:

**Reward Phasing (RP).** The *Reward Phasing* algorithm (Bajaj et al., 2023) is a recent IL-to-RL transfer method that explicitly phases out the imitation signal which is designed as an IRL Arora & Doshi (2021)

Table 4: Learning performance of CAIR (BC+SAC and DAgger+DDPG) compared to hybrid IL–RL baselines on D4RL locomotion tasks. Values report final reward (mean ± standard deviation), AUC, and normalized reward (NormReward), computed as $(R - R_{\min})/(R_{\text{demo}} - R_{\min})$, where $R_{\min}$ is the minimum score observed across all evaluated methods; values above 1 indicate performance exceeding the demonstrator. Methods marked with † are significantly worse than the highest-mean method in the same task (Welch's t-test, $p < 0.05$). Among the remaining methods, AUC determines the highest-ranked method, which is bolded. The final rows report average NormReward across tasks.

| Task | Algorithm | Mean | ± SD | AUC | NormReward |
|---|---|---|---|---|---|
| *HalfCheetah* (250k steps) | **DAgger+DDPG** | **9,584** | **± 1,119** | **6,649** | **0.96** |
| | BC+SAC | †7,113 | ± 332 | 3,677 | 0.73 |
| | RP | †848 | ± 112 | 439 | 0.13 |
| | SAIL | †-557 | ± 91 | -537 | 0.00 |
| | JumpStart | †3,502 | ± 1,575 | 1,914 | 0.38 |
| | RLPD | †1,037 | ± 662 | 935 | 0.15 |
| *Hopper* (250k steps) | DAgger+DDPG | †3,328 | ± 210 | 2,944 | 0.99 |
| | BC+SAC | †3,424 | ± 119 | 2,864 | 1.03 |
| | RP | †2,704 | ± 146 | 2,344 | 0.73 |
| | SAIL | †1,083 | ± 306 | 1,425 | 0.05 |
| | JumpStart | †976 | ± 38 | 978 | 0.00 |
| | **RLPD** | **3,705** | **± 141** | **3,288** | **1.15** |
| *Walker2d* (250k steps) | DAgger+DDPG | †3,369 | ± 1,044 | 3,072 | 0.83 |
| | BC+SAC | †3,814 | ± 325 | 2,534 | 0.93 |
| | RP | †1,609 | ± 420 | 1,395 | 0.42 |
| | SAIL | †-187 | ± 1,002 | 266 | 0.00 |
| | JumpStart | †827 | ± 14 | 826 | 0.24 |
| | **RLPD** | **5,232** | **± 213** | **4,241** | **1.26** |
| *Swimmer* (250k steps) | **DAgger+DDPG** | **306.1** | **± 68.5** | **289.3** | **0.93** |
| | BC+SAC | 256.8 | ± 22.2 | 153.6 | 0.78 |
| | RP | †0.1 | ± 0.2 | 1.2 | 0.01 |
| | SAIL | †-0.2 | ± 1.1 | -0.2 | 0.00 |
| | JumpStart | †5.8 | ± 20.2 | 4.2 | 0.02 |
| | RLPD | †2.9 | ± 1.1 | 3.1 | 0.01 |
| *Ant* (250k steps) | DAgger+DDPG | †1,275 | ± 672 | 1,550 | 0.67 |
| | **BC+SAC** | **3,666** | **± 902** | **1,853** | **1.04** |
| | RP | †-301 | ± 21 | -471 | 0.42 |
| | SAIL | †-3,001 | ± 16 | -2,743 | 0.00 |
| | JumpStart | †710 | ± 454 | 643 | 0.58 |
| | RLPD | †654 | ± 15 | 471 | 0.57 |
| *Humanoid* (250k steps) | DAgger+DDPG | †5,346 | ± 97 | 5,248 | 1.09 |
| | BC+SAC | †5,024 | ± 171 | 4,934 | 0.79 |
| | RP | †4,177 | ± 100 | 4,134 | 0.00 |
| | SAIL | †4,392 | ± 392 | 4,368 | 0.30 |
| | JumpStart | †4,927 | ± 10 | 4,927 | 0.70 |
| | **RLPD** | **6,386** | **± 258** | **5,765** | **2.06** |
| *Average Normalized Reward Across Tasks* | **DAgger+DDPG** | | | | **0.91** |
| | BC+SAC | | | | 0.88 |
| | RP | | | | 0.28 |
| | SAIL | | | | 0.04 |
| | JumpStart | | | | 0.32 |
| | RLPD | | | | 0.87 |

approximated reward function. It has demonstrated state-of-the-art performance in the standard Fetch tasks (Plappert et al., 2018).

**Hindsight Experience Replay (HER) with Demonstrations.** Hindsight Experience Replay (Andrychowicz et al., 2017) addresses sparse-reward learning by relabeling failed trajectories as successes

with respect to the achieved goal. We compare against the extension proposed by Nair et al. (Nair et al., 2018), which explicitly incorporates demonstrations by seeding the replay buffer with demonstration data and maintaining a fixed ratio of demonstration transitions during mini-batch sampling. This guides exploration toward relevant regions of the state space. Since *HER* relies on goal relabeling, we evaluate this baseline only on the goal-conditioned manipulation tasks and exclude it from the locomotion experiments.

**Self-Adaptive Imitation Learning (SAIL).** SAIL (Zhu et al., 2022) is an off-policy algorithm that dynamically expands the teacher's demonstration buffer with high-quality, self-generated trajectories. The approach effectively optimizes a convex combination of Behavioral Cloning and Q-learning objectives, where the weight is adapted based on the estimated quality of the generated data.

**JumpStart Reinforcement Learning (JSRL).** JSRL (Uchendu et al., 2022) is a curriculum-based meta-algorithm that initializes the RL policy using a pre-trained guide (the demonstrator). It progressively increases the difficulty of the task by initializing the agent at states further from the goal (using the demonstrator to reach those states), thereby creating a curriculum of starting distributions that gradually transitions from the demonstrator's capabilities to the RL agent's control.

**Reinforcement Learning with Prior Data (RLPD).** RLPD (Ball et al., 2023) is a hybrid method designed for sample efficiency. It combines offline demonstrations with online interactions using a high Update-to-Data (UTD) ratio, layer normalization, and an ensemble of critics. Unlike other methods that rely on explicit imitation losses, RLPD biases the learning through symmetric sampling of demonstration data and aggressive gradient updates.

Further details on these methods and their specific hyperparameters are provided in Appendix A.5 and Appendix A.2, respectively.

### 4.4 Sample Efficiency and Asymptotic Performance Results

To evaluate the effectiveness of CAIR, we compare representative CAIR variants against the baselines. For the manipulation tasks, we report BC+DDPG and BC+SAC, which achieve the strongest overall performance among the evaluated manipulation pairings. For the locomotion tasks, we report BC+SAC and DAgger+DDPG. Although DAgger+SAC achieves the highest average normalized reward across the locomotion benchmark tasks, BC+SAC is included to enable a more direct comparison with prior hybrid IL–RL baselines, which primarily rely on offline demonstrations rather than interactive demonstrators.

**Manipulation Tasks.** Table 3 summarizes the final performance and area under the learning curve (AUC) achieved by CAIR and the evaluated baselines on the manipulation tasks. The corresponding learning curves are provided in Figure 7 in Appendix A.4. Relative to the evaluated baselines, CAIR achieves the highest-ranked performance in three of the five manipulation tasks, namely *FetchPickPlace-v1*, *FetchPush-v1*, and *FetchSlide-v1*. On *AdroitHandHammerSparse-v1*, BC+SAC and RLPD have statistically comparable final return, with BC+SAC having the higher mean final return and RLPD the higher AUC. On *AdroitHandPenSparse-v1*, BC+SAC achieves performance comparable to RLPD, with both methods substantially outperforming the remaining baselines. Consistent with these task-level results, BC+SAC achieves the highest average normalized reward across tasks (8.23), exceeding both RLPD (7.00) and the remaining baselines. Across all five manipulation tasks, CAIR consistently learns policies that match or exceed demonstrator performance.

**Locomotion Tasks.** Table 4 summarizes the final performance and area under the learning curve (AUC) achieved by CAIR and the evaluated baselines on the D4RL locomotion tasks. The corresponding learning curves are provided in Figure 8 in Appendix A.4. Relative to the evaluated baselines, CAIR achieves the highest-ranked performance in three of the six locomotion tasks (*HalfCheetah-v5*, *Swimmer-v5*, and *Ant-v5*). On *Hopper-v5*, BC+SAC achieves performance comparable to RLPD, although RLPD achieves the stronger overall result. RLPD also achieves the strongest overall performance on *Walker2d-v5* and *Humanoid-v5*. As a complementary aggregate summary, DAgger+DDPG attains the highest average normalized reward across tasks (0.91), marginally exceeding both BC+SAC (0.88) and RLPD (0.87).

### 4.5 Ablation Study: Impact of CAGrad

To address Research Question 3, we investigate the contribution of Conflict-Averse Gradient Descent (CAGrad) beyond loss annealing. Specifically, we compare variants with and without CAGrad under matched annealing configurations. Within each comparison, the underlying IL–RL learner, annealing type, and, for fixed schedules, the step size $\Delta$ are held constant. The analysis includes both the proposed adaptive schedule and fixed schedules with $\Delta \in \{0.01, 0.05, 0.1, 0.15\}$. This isolates the effect of conflict-aware gradient manipulation for a given annealing configuration while also evaluating whether its effect is consistent across annealing schedules. At each recorded update, we define the worst-case cosine alignment as $c_{\mathrm{worst}} = \min\{\cos(d, g_{\mathrm{IL}}), \cos(d, g_{\mathrm{RL}})\}$, where $d$ is the final update direction. This quantity measures alignment with the less-aligned of the two objectives. For each configuration, the reported mean worst-case cosine is averaged over its recorded gradient updates. For a matched comparison, we report

$$\Delta c_{\mathrm{worst}} = c_{\mathrm{worst}}^{+\mathrm{CAGrad}} - c_{\mathrm{worst}}^{\mathrm{No\ CAGrad}}$$

so positive values indicate improved alignment with the less-aligned objective gradient.

The learning-curve ablations include DAgger+DDPG and DAgger+SAC, while Tables 13 and 15 report controlled matched comparisons for DAgger+DDPG. Final reward is compared using a two-sided Welch's $t$-test. When one configuration is significantly worse, its matched alternative is treated as favored. When the test does not distinguish the two configurations, they are treated as statistically comparable, and AUC is used only to determine which configuration is favored for the summary counts.

**Manipulation Tasks.** The manipulation results are presented in Figure 9 and Table 13. Across the 15 matched adaptive and fixed-schedule comparisons, incorporating CAGrad is favored under this evaluation rule in 11 configurations, statistically comparable but not AUC-favored in two, and significantly worse in two. Thus, CAGrad is favored or statistically comparable in 13 of 15 comparisons.

The mean change in worst-case cosine is positive for each Fetch task when aggregated across its five annealing configurations, with an overall paired increase of $0.072 \pm 0.070$. This indicates that CAGrad generally shifts the joint update toward better alignment with the less-aligned objective gradient. However, the two configurations in which CAGrad obtains lower reward despite a positive cosine improvement show that improved worst-case alignment does not necessarily translate into improved policy return.

**Locomotion Tasks.** Figure 10 and Table 15 present the corresponding analysis on the D4RL locomotion tasks. Across 30 matched adaptive and fixed-schedule comparisons, CAGrad is favored under this evaluation rule in 15 configurations, statistically comparable but not AUC-favored in five, and significantly worse in ten. It is therefore favored or statistically comparable in 20 of 30 comparisons.

CAGrad improves mean worst-case cosine in all 30 locomotion comparisons, with an average paired increase of $0.202 \pm 0.133$. This provides consistent evidence that CAGrad alters the update in the intended conflict-averse direction. Nevertheless, its less uniform effect on final reward and AUC shows that improved gradient alignment alone is not sufficient to guarantee better learning performance for every task and annealing schedule.

**Overall Assessment.** Overall, the ablation results provide qualified support for Research Question 3. CAGrad consistently improves the measured worst-case gradient alignment and is frequently favored or statistically comparable in terms of reward. Its performance benefit is nevertheless task- and schedule-dependent. We therefore interpret CAGrad as an effective mechanism for mitigating IL–RL gradient conflict, rather than as a guarantee of improved sample efficiency or asymptotic performance in every configuration.

### 4.6 Ablation Study: Impact of Adaptive Annealing

To study the effect of the proposed adaptive annealing strategy, we compare CAIR against fixed-step annealing variants that use the same CAGrad update, with $\lambda \leftarrow \lambda - \Delta$ for $\Delta \in \{0.01, 0.05, 0.1, 0.15\}$. This isolates the benefit of combining adaptive annealing with CAGrad relative to using CAGrad with tuned fixed schedules, while the complementary CAGrad ablation tables compare adaptive annealing with and without CAGrad. Final reward is compared using a two-sided Welch's $t$-test. Schedules significantly worse than the highest-mean

schedule are treated as underperforming; among the remaining statistically comparable schedules, AUC determines which schedule is the most favored.

**Manipulation Tasks.** Figure 14 and Table 12 show that adaptive annealing is favored under this evaluation rule in one of the three Fetch tasks and is statistically comparable, but not AUC-favored, in the remaining two. It is not significantly worse in any manipulation task. The favored schedule varies across tasks: adaptive annealing, $\Delta = 0.01$, and $\Delta = 0.05$ are each favored once, while neither $\Delta = 0.1$ nor $\Delta = 0.15$ is favored. More aggressive schedules also degrade performance in several cases, particularly at $\Delta = 0.15$.

These results indicate that no single fixed annealing rate is consistently favored across the Fetch tasks. Adaptive annealing therefore avoids the need to choose $\Delta$ separately for each task while remaining favored or statistically comparable to the strongest tested fixed schedule in all three tasks.

**Locomotion Tasks.** Figure 15 and Table 14 present the corresponding analysis on the D4RL locomotion tasks. Adaptive annealing is favored under this evaluation rule in two of six tasks, is statistically comparable but not AUC-favored in two others, and is significantly worse than the highest-mean fixed schedule in the remaining two. Among the four tasks in which a fixed schedule is favored, $\Delta = 0.01$ and $\Delta = 0.05$ are each favored twice, whereas neither $\Delta = 0.1$ nor $\Delta = 0.15$ is favored.

The learning curves further show that larger fixed rates can remove imitation guidance too quickly, producing a pronounced decline as $\lambda$ approaches zero. Adaptive annealing does not eliminate such failures in every task, but it remains favored or statistically comparable in four of six domains without requiring a task-specific choice of $\Delta$.

**Overall Assessment.** Across the nine manipulation and locomotion tasks, adaptive annealing is favored in three, statistically comparable but not AUC-favored in four, and significantly worse in two. Among the six tasks in which a fixed schedule is favored, $\Delta = 0.01$ is favored three times and $\Delta = 0.05$ three times, while the larger rates are never favored. Thus, the results do not support the claim that adaptive annealing uniformly outperforms an oracle-tuned fixed schedule. Instead, they show that it provides a competitive, data-dependent alternative that avoids the need to choose a fixed annealing rate separately for each task.

## 5 Conclusions

We introduced *Conflict-Averse IL-RL* (CAIR), a framework for combining Imitation Learning (IL) and Reinforcement Learning (RL) while addressing gradient conflict as one potential contributor to negative transfer during the IL-to-RL transition. CAIR adaptively adjusts the relative weighting of the IL and RL objectives using their gradient alignment and incorporates Conflict-Averse Gradient Descent (CAGrad) to improve the worst-case alignment of the resulting update with both objectives. Together, these mechanisms are designed to mitigate harmful gradient interference as optimization shifts from imitation-driven initialization to reward-driven learning.

CAIR admits a conditional monotonic non-deterioration result for the idealized stagewise policy sequence satisfying the stated KL-proximity and minimum surrogate-improvement assumptions. CAIR does not itself enforce these assumptions, and the formal result therefore does not extend directly to the SAC and DDPG implementations. Empirically, we evaluate CAIR with PPO, SAC, and DDPG across five sparse-reward manipulation tasks and six dense-reward locomotion tasks. Relative to the evaluated hybrid IL–RL baselines, at least one reported CAIR variant achieves the highest-ranked result in three of five manipulation tasks and three of six locomotion tasks. CAIR is competitive with the strongest baseline on the remaining manipulation tasks and on *Hopper-v5*, while RLPD performs better on *Walker2d-v5* and *Humanoid-v5*. These results show that CAIR is competitive across both benchmark suites, although its relative performance remains task- and pairing-dependent.

Our ablation studies clarify the roles of the two CAIR components. CAGrad improves the measured worst-case gradient alignment, but its effect on final return and AUC depends on the task and annealing schedule, showing that improved alignment alone does not guarantee better policy performance. Adaptive annealing provides a competitive, data-dependent alternative to fixed schedules by avoiding task-specific selection of a

fixed annealing rate while remaining favored or statistically comparable to the strongest tested fixed schedule in the majority of the evaluated tasks.

A practical limitation of CAIR is its sensitivity to the compatibility of the underlying IL and RL algorithms. Performance varies substantially across IL–RL pairings, and no single pairing consistently achieves the strongest result across all tasks. Pairing selection therefore remains domain-dependent. Future work will investigate methods for characterizing and improving compatibility across IL–RL pairings and extend CAIR to multimodal demonstration data.

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

# A   Appendix

## A.1   Task Descriptions

We evaluate CAIR on two continuous control benchmark suites categorized by sparse reward manipulation tasks (Fetch and Adroit Tasks) and dense-reward locomotion tasks (D4RL MujoCo tasks).

### A.1.1   Fetch Robotics Tasks

We use three tasks based on the 7-DoF Fetch robotics arm (Plappert et al., 2018): *FetchPickAndPlace-v1* (P&P), *FetchSlide-v1* (FS), and *FetchPush-v1* (FP). Visualizations are provided in Figure 4. These are multi-goal tasks where the target position is randomized every episode. A rule-based suboptimal demonstrator (Bajaj et al., 2023) is used for each task.

**Common Properties:**

- **State Space $\mathcal{S}$:** Observations include the Cartesian position, linear velocity, and gripper state of the robot, along with the object's position, rotation (Euler angles), velocities, and relative position to the gripper. The desired target coordinates and achieved goals are also observed (required for HER).

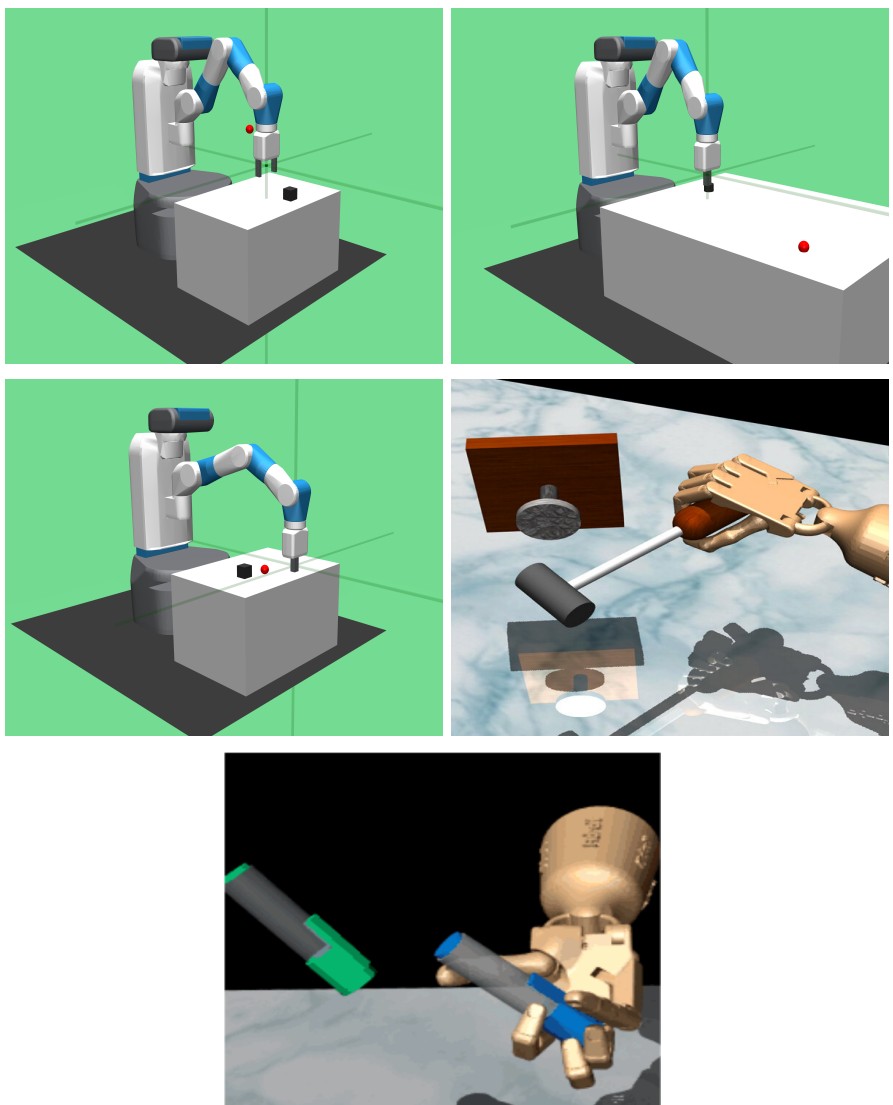

Figure 4: Visualizations of the five MuJoCo sparse-reward tasks used in our experiments. Top-left: FetchPickAndPlace-v1 task; Top-right: FetchSlide-v1 task; Middle-left: FetchPush-v1 task; Middle-right: AdroitHandHammerSparse-v1; Bottom: AdroitHandPenSparse-v1

- **Action Space** $\mathcal{A}$**:** 4-dimensional continuous space. 3 dimensions control gripper movement in Cartesian coordinates ($\in [-1, 1]$), and the last dimension controls the gripper state ($\in [-1, 1]$). In *FetchSlide* and *FetchPush*, the gripper is locked.

- **Transition Function** $\mathcal{P}$**:** The robot updates its position based on the action. Each step corresponds to 20 simulator steps ($\Delta t = 0.002s$).

- **Reward Function** $R_T$**:** Sparse binary reward. 0 if the object is within 5cm of the target, $-1$ otherwise.

**Task Specifics:**

- **FetchPickAndPlace-v1:** The robot must grasp a box and move it to a target in the air or on the table. This is the most complex Fetch task due to the grasping requirement.

- **FetchSlide-v1:** The robot must push a puck across a long table to a target outside its immediate reach. This requires learning the object's sliding dynamics to apply precise force.

- **FetchPush-v1:** The robot pushes a box to a target location within its reach. This is simpler than Slide but still requires precise manipulation.

### A.1.2  Adroit Manipulation Tasks

We evaluate CAIR on two dexterous manipulation tasks from the D4RL benchmark (Fu et al., 2020): *AdroitHandHammerSparse-v1* (Hammer) and *AdroitPenSparse-v1* (Pen). Visualizations are provided in Figure 4. Both tasks use the 30-DoF Shadow Hand to perform fine-grained manipulation tasks requiring coordinated finger control. Expert demonstrations for these tasks are obtained from the D4RL datasets.

**Common Properties:**

- **State Space $\mathcal{S}$:** Observations include the positions, orientations, and velocities of manipulated objects, along with the Shadow Hand joint angles and joint velocities.

- **Action Space $\mathcal{A}$:** 24-dimensional continuous action space representing joint torques of the Shadow Hand ($\in [-1, 1]$).

- **Transition Function $\mathcal{P}$:** Actions apply torques to the hand joints, producing object manipulation dynamics governed by MuJoCo physics.

**Task Specifics:**

- **AdroitHandHammerSparse-v1:** The hand must grasp a hammer and drive a nail into a board. The state includes hammer pose, nail position, and nail depth. The reward is sparse: $+10$ when the nail is fully driven and $-0.1$ otherwise to encourage efficiency.

- **AdroitHandPenSparse-v1:** The hand must reorient a pen to match a target orientation. Observations include the pen pose, angular velocities, and target orientation. The reward is sparse: $+50$ when the pen orientation is close.

### A.1.3  D4RL Locomotion Tasks

We additionally evaluate CAIR on six continuous locomotion tasks from the D4RL benchmark (Fu et al., 2020): *HalfCheetah*, *Hopper*, *Walker2d*, *Ant*, *Swimmer*, and *Humanoid*. These environments involve high-dimensional continuous control tasks where the objective is to maximize forward locomotion reward under complex MuJoCo dynamics. Unlike the sparse-reward manipulation tasks, these tasks use dense reward functions and evaluate sustained locomotion behavior.

**Common Properties:**

- **State Space $\mathcal{S}$:** Observations consist of joint positions, joint velocities, body orientations, and task-specific proprioceptive features describing the current locomotion state.

- **Action Space $\mathcal{A}$:** Continuous action spaces corresponding to actuator torques applied to the robot joints.

- **Transition Function $\mathcal{P}$:** Actions apply torques to the articulated body, with future states determined by MuJoCo physics simulation.

- **Reward Function $R_T$:** Dense reward functions encouraging forward velocity while penalizing unstable motion and excessive control effort.

**Task Specifics:**

- **HalfCheetah:** A planar two-legged agent that learns fast forward locomotion through coordinated joint movement.

- **Hopper:** A single-legged hopping robot that must balance while maintaining forward movement.

- **Walker2d:** A bipedal robot that must learn stable walking and running behavior while avoiding falls.

- **Ant:** A four-legged quadrupedal robot with a high-dimensional action space requiring coordinated locomotion control.

- **Swimmer:** A low-dimensional swimming agent that learns propulsion through oscillatory body motion.

- **Humanoid:** A high-dimensional humanoid robot requiring coordinated full-body locomotion and balance.

### A.2    Hyperparameters

Experiments were conducted on an AMD Ryzen Threadripper PRO 5975WX (32-Core, 4.3GHz). Tables 5–7 and Tables 8–10 detail the hyperparameters used across the manipulation and locomotion tasks, respectively. Unless otherwise specified, all network architectures use tanh activations.

**Manipulation Tasks:** For the Fetch and Adroit environments, we use 50 demonstration trajectories per task. Episode lengths are capped at 1,000 steps for the Fetch tasks and 4,000 steps for the Adroit tasks.

**Locomotion Tasks:** For the D4RL locomotion environments, we use the corresponding `simple` Minari datasets (Younis et al., 2024). Hyperparameters are initialized using the default RL Baselines3 Zoo benchmark configurations (Raffin, 2020), with only minor modifications where required for stable IL–RL annealing.

| Task | Network | $\gamma$ | LR | Batch | $\tau$ | Ent Coef |
|---|---|---|---|---|---|---|
| P&P, Slide | (512, 512) | 0.98 | $3 \cdot 10^{-4}$ | 128 | 0.02 | Auto |
| Push | (512, 512) | 0.98 | $3 \cdot 10^{-5}$ | 256 | 0.02 | Auto |
| AdroitHammer/Pen | (512, 512) | 0.98 | $3 \cdot 10^{-4}$ | 256 | 0.02 | Auto |

Table 5: Hyperparameters for CAIR (BC/DAgger + SAC)

| Task | Network | $\gamma$ | LR | Batch | $\tau$ | Noise $\sigma$ |
|---|---|---|---|---|---|---|
| P&P, Slide | (512, 512) | 0.98 | $3 \cdot 10^{-4}$ | 128 | 0.02 | 0.15 |
| Push | (512, 512) | 0.98 | $3 \cdot 10^{-5}$ | 256 | 0.02 | 0.15 |
| AdroitHammer/Pen | (512, 512) | 0.98 | $3 \cdot 10^{-4}$ | 128 | 0.02 | 0.10 |

Table 6: Hyperparameters for CAIR (BC/DAgger + DDPG)

| Task | Network | $\gamma$ | Init $\lambda$ | LR | Rollout Steps | Epochs | Ent Coef |
|---|---|---|---|---|---|---|---|
| P&P | (512, 512) | 0.98 | 0.95 | $3 \cdot 10^{-4}$ | 1024 | 30 | 0.2 |
| Slide, Push | (512, 512) | 0.98 | 0.95 | $3 \cdot 10^{-4}$ | 1024 | 20 | 0.02 |
| AdroitHammer/Pen | (512, 512) | 0.98 | 0.95 | $3 \cdot 10^{-4}$ | 1024 | 20 | 0.02 |

Table 7: Hyperparameters for CAIR (BC/DAgger + PPO)

### A.3    Sensitivity Analysis

We provide additional empirical analysis of the CAIR framework components.

- **CAGrad Ablation (Fig. 9):** Evaluates how CAGrad affects gradient alignment and return under matched annealing configurations.

| Task | Network | $\gamma$ | LR | Batch | $\tau$ | Ent Coef |
|------|---------|----------|-----|-------|--------|----------|
| HalfCheetah | (256, 256) | 0.99 | $3 \cdot 10^{-4}$ | 256 | 0.005 | Auto |
| Hopper | (256, 256) | 0.99 | $3 \cdot 10^{-4}$ | 256 | 0.005 | Auto |
| Walker2d | (256, 256) | 0.99 | $3 \cdot 10^{-4}$ | 256 | 0.005 | Auto |
| Ant | (256, 256) | 0.99 | $3 \cdot 10^{-4}$ | 256 | 0.005 | Auto |
| Swimmer | (256, 256) | 0.999 | $3 \cdot 10^{-4}$ | 256 | 0.005 | Auto |
| Humanoid | (256, 256) | 0.99 | $3 \cdot 10^{-4}$ | 256 | 0.005 | Auto |

Table 8: Hyperparameters for CAIR (BC/DAgger + SAC) on locomotion tasks

| Task | Network | $\gamma$ | LR | Batch | $\tau$ | Noise $\sigma$ |
|------|---------|----------|-----|-------|--------|----------------|
| HalfCheetah | (256, 256) | 0.99 | $1 \cdot 10^{-3}$ | 256 | 0.005 | 0.10 |
| Hopper | (256, 256) | 0.99 | $1 \cdot 10^{-3}$ | 256 | 0.005 | 0.10 |
| Walker2d | (256, 256) | 0.99 | $1 \cdot 10^{-3}$ | 256 | 0.005 | 0.10 |
| Ant | (256, 256) | 0.99 | $1 \cdot 10^{-3}$ | 256 | 0.005 | 0.10 |
| Swimmer | (256, 256) | 0.9999 | $1 \cdot 10^{-3}$ | 256 | 0.005 | 0.10 |
| Humanoid | (256, 256) | 0.99 | $1 \cdot 10^{-3}$ | 256 | 0.005 | 0.10 |

Table 9: Hyperparameters for CAIR (BC/DAgger + DDPG) on locomotion tasks

| Task | Network | $\gamma$ | Init $\lambda$ | LR | Rollout Steps | Epochs | Ent Coef | Clip Range |
|------|---------|----------|---------------|-----|---------------|--------|----------|------------|
| HalfCheetah | (256, 256) | 0.98 | 0.92 | $2.06 \cdot 10^{-5}$ | 512 | 20 | 0.000401 | 0.1 |
| Hopper | (256, 256) | 0.999 | 0.99 | $9.808 \cdot 10^{-5}$ | 512 | 5 | 0.002295 | 0.2 |
| Walker2d | (256, 256) | 0.99 | 0.95 | $5.05 \cdot 10^{-5}$ | 512 | 20 | 0.000585 | 0.1 |
| Ant | (256, 256) | 0.99 | 0.95 | $3 \cdot 10^{-4}$ | 2048 | 10 | 0.0 | 0.2 |
| Swimmer | (256, 256) | 0.9999 | 0.98 | $3 \cdot 10^{-4}$ | 1024 | 10 | 0.0 | 0.2 |
| Humanoid | (256, 256) | 0.95 | 0.9 | $3.569 \cdot 10^{-5}$ | 512 | 5 | 0.00238 | 0.3 |

Table 10: Hyperparameters for CAIR (BC/DAgger + PPO) on locomotion tasks

- **Sensitivity to $\eta_\lambda$ (Fig. 11):** Shows that extreme learning rates for the $\lambda$-optimizer destabilize learning.

- **Update Interval (Fig. 12):** Highlights the trade-off between responsiveness and stability in $\lambda$ updates.

## A.4 Conflict-Averse IL to RL Algorithm

CAIR optimizes a convex combination of imitation and reinforcement learning losses, $L_\lambda = (1 - \lambda)L_{\mathrm{RL}} + \lambda L_{\mathrm{IL}}$, while mitigating gradient conflicts via CAGrad and adaptive optimization of the mixture coefficient $\lambda$.

We evaluate CAIR with SAC, DDPG, and PPO as the RL algorithms, combined with Behavior Cloning (BC) and DAgger as the IL algorithms. The corresponding hyperparameters are reported in Tables 5, 6, 7, 8, 9, and 10. All remaining hyperparameters follow the default settings of Stable-Baselines3 (Raffin et al., 2021). The constraint parameter $c$ for CAGrad and the learning rate for the $\lambda$-optimizer are fixed across tasks unless stated otherwise. The adaptive $\lambda$-optimization procedure is summarized in Algorithm 2 (Appendix A.4).

## A.5 Baseline Implementation Details

### A.5.1 Reward Phasing (RP)

Reward Phasing uses demonstrations to learn an auxiliary reward function via Adversarial Inverse Reinforcement Learning (AIRL) (Fu et al., 2017), which is then annealed during RL training. We follow the formulation and hyperparameters from Bajaj et al. (Bajaj et al., 2023). Annealing is performed every 200

---

**Algorithm 1** Conflict-Averse IL to RL (CAIR)

---

1: **Input**: Policy Parameters $\theta_0$, CAGrad constraint $c \in [0, 1)$, Demonstrations $D$, Optimizer `Opt`, Task update frequency $T_\lambda$.
2: Initialize $\lambda \leftarrow 1.0$, $\theta \leftarrow \theta_0$, $k \leftarrow 1$, $\mathcal{B} \leftarrow \emptyset$
3: **for** each episode **do**
4:     **for** each step $t$ **do**
5:         $a_t \sim \pi(a_t \mid s_t; \theta_k)$
6:         Execute $a_t$, observe $s_{t+1}, r_t$
7:         $\mathcal{B} \leftarrow \mathcal{B} \cup \{(s_t, a_t, r_t, s_{t+1})\}$
8:     **end for**
9:     **for** each gradient step **do**
10:       **if** Annealing Complete ($\lambda = 0$) **then**
11:         Compute gradients $g_{RL}$
12:         $\theta_{k+1} \leftarrow \mathtt{Opt}(\theta_k, g_{RL})$
13:       **else**
14:         $L_\lambda \leftarrow (1 - \lambda)L_{\text{RL}}(\theta, \mathcal{B}) + \lambda L_{\text{IL}}(\theta, D)$
15:         Compute gradients $g_{IL}, g_{RL}, g_\lambda$
16:         $d = \mathbf{CAGrad}(g_\lambda, [g_{IL}, g_{RL}], c)$
17:         $\theta_{k+1} \leftarrow \mathtt{Opt}(\theta_k, d)$
18:       **end if**
19:       $k \leftarrow k + 1$
20:     **end for**
21:     **if** $\lambda > 0$ **then**
22:       Every $T_\lambda$ episodes do $\lambda \leftarrow \textsc{OptimizeLambda}(g_{\text{IL}}, g_{\text{RL}}, \lambda)$
23:     **end if**
24: **end for**

---

**Algorithm 2** $\textsc{OptimizeLambda}(g_{\text{IL}}, g_{\text{RL}}, \lambda_{\text{cur}})$

---

1: **Input:** Gradients $g_{\text{IL}}, g_{\text{RL}}$, current $\lambda_{\text{cur}}$, learning rate $\eta_\lambda$, threshold $\epsilon_\lambda > 0$
2: Normalize: $\tilde{g}_{\text{IL}} \leftarrow g_{\text{IL}}/\|g_{\text{IL}}\|$, $\tilde{g}_{\text{RL}} \leftarrow g_{\text{RL}}/\|g_{\text{RL}}\|$
3: Initialize primal $\lambda \leftarrow \lambda_{\text{cur}}$, dual $\mu \leftarrow \mu_0$
4: **for** $k = 1$ to $K$ **do**
5:     $\tilde{g}_\lambda \leftarrow (1 - \lambda)\tilde{g}_{\text{RL}} + \lambda\tilde{g}_{\text{IL}}$
6:     $c_{\text{RL}} \leftarrow \langle \tilde{g}_{\text{RL}}, \tilde{g}_\lambda \rangle$, $c_{\text{IL}} \leftarrow \langle \tilde{g}_{\text{IL}}, \tilde{g}_\lambda \rangle$ {Cosine similarity between gradients}
7:     $J(\lambda, \mu) \leftarrow c_{\text{RL}} - \mu \max(0, -c_{\text{IL}})$ {RL alignment objective with IL misalignment penalty}
8:     $\lambda \leftarrow \lambda + \eta_\lambda \nabla_\lambda J$ {Gradient Ascent}
9:     $\lambda \leftarrow clamp(\lambda, 0, 1)$ {Project to $[0, 1]$}
10:     $\mu \leftarrow \max(0, \mu + \eta_\mu \max(0, -c_{\text{IL}}))$ {Dual Update}
11:     **if** $\lambda < \epsilon_\lambda$ **then**
12:       $\lambda \leftarrow 0$ {Annealing Completion Condition}
13:     **end if**
14: **end for**
15: **return** $\lambda$

---

Table 11: Task-specific reference scores used to compute normalized reward, $\text{NormReward} = \frac{R - R_{\min}}{R_{\text{demo}} - R_{\min}}$. Here, $R_{\min}$ denotes the task-specific normalization reference and $R_{\text{demo}}$ denotes the corresponding demonstrator return.

| Suite | Task | $R_{\min}$ | $R_{\text{demo}}$ |
|---|---|---|---|
| Manipulation | *FetchPickAndPlace-v1* | -50 | -35 |
| | *FetchPush-v1* | -50 | -30 |
| | *FetchSlide-v1* | -50 | -45 |
| | *AdroitHandHammerSparse-v1* | -20 | 26 |
| | *AdroitHandPenSparse-v1* | 0 | 3,500 |
| Locomotion | *HalfCheetah-v5* | -557 | 10,000 |
| | *Hopper-v5* | 976 | 3,350 |
| | *Walker2d-v5* | -187 | 4,100 |
| | *Swimmer-v5* | -0.2 | 330 |
| | *Ant-v5* | -3,001 | 3,400 |
| | *Humanoid-v5* | 4,177 | 5,250 |

Table 12: Adaptive-annealing ablation on Fetch manipulation tasks with DAgger+DDPG and CAGrad enabled in all variants. Values report final reward (mean $\pm$ standard deviation) and area under the learning curve over the first 8,000 episodes (AUC@8k). Methods marked with † are significantly worse than the highest-mean schedule for the same task according to a two-sided Welch's $t$-test ($p < 0.05$; $n = 5$ per configuration). Among the remaining schedules, the method with the highest AUC@8k is bolded.

| Task | Annealing | $\delta$ | Mean | $\pm$ SD | AUC@8k |
|---|---|---|---|---|---|
| *FetchPickAndPlace* (8k episodes) | **Adaptive** | − | **-20.11** | **$\pm$ 12.20** | **-30.52** |
| | Fixed | 0.01 | -16.96 | $\pm$ 13.13 | -31.81 |
| | Fixed | 0.05 | -28.87 | $\pm$ 27.18 | -39.07 |
| | Fixed | 0.10 | -36.06 | $\pm$ 17.45 | -45.99 |
| | Fixed | 0.15 | -35.70 | $\pm$ 17.15 | -46.33 |
| *FetchPush* (8k episodes) | Adaptive | − | -10.70 | $\pm$ 2.24 | -20.22 |
| | **Fixed** | **0.01** | **-9.49** | **$\pm$ 1.07** | **-16.17** |
| | Fixed | 0.05 | -9.72 | $\pm$ 1.22 | -18.89 |
| | Fixed | 0.10 | -9.91 | $\pm$ 1.25 | -21.81 |
| | Fixed | 0.15 | †-46.42 | $\pm$ 2.19 | -46.61 |
| *FetchSlide* (8k episodes) | Adaptive | − | -36.54 | $\pm$ 5.34 | -37.73 |
| | Fixed | 0.01 | -38.81 | $\pm$ 5.38 | -43.79 |
| | **Fixed** | **0.05** | **-32.94** | **$\pm$ 1.85** | **-35.63** |
| | Fixed | 0.10 | †-49.35 | $\pm$ 0.00 | -48.18 |
| | Fixed | 0.15 | †-44.87 | $\pm$ 3.99 | -48.49 |

training episodes for Fetch tasks and every 20 episodes for AdroitHandHammer. To prevent performance collapse, the entropy coefficient is annealed from 0.005 to 0.001, the learning rate from $3 \times 10^{-4}$ to $7 \times 10^{-5}$, and the AIRL reward weight $\alpha$ to 0.001 after 75% of the phasing schedule.

### A.5.2 Hindsight Experience Replay (HER) with Demonstrations

HER with demonstrations augments the replay buffer with demonstration trajectories to reduce exploration difficulty in sparse-reward environments. We use DDPG (Lillicrap et al., 2015) as the underlying RL algorithm. The HER buffer is initialized with demonstration data. All remaining hyperparameters follow the default HER implementation.

### A.5.3 Self-Adaptive Imitation Learning (SAIL)

SAIL dynamically augments the demonstration buffer with high-quality self-generated trajectories during training. We use TD3 (Fujimoto et al., 2018) as the policy optimization algorithm. All hyperparameters, net-

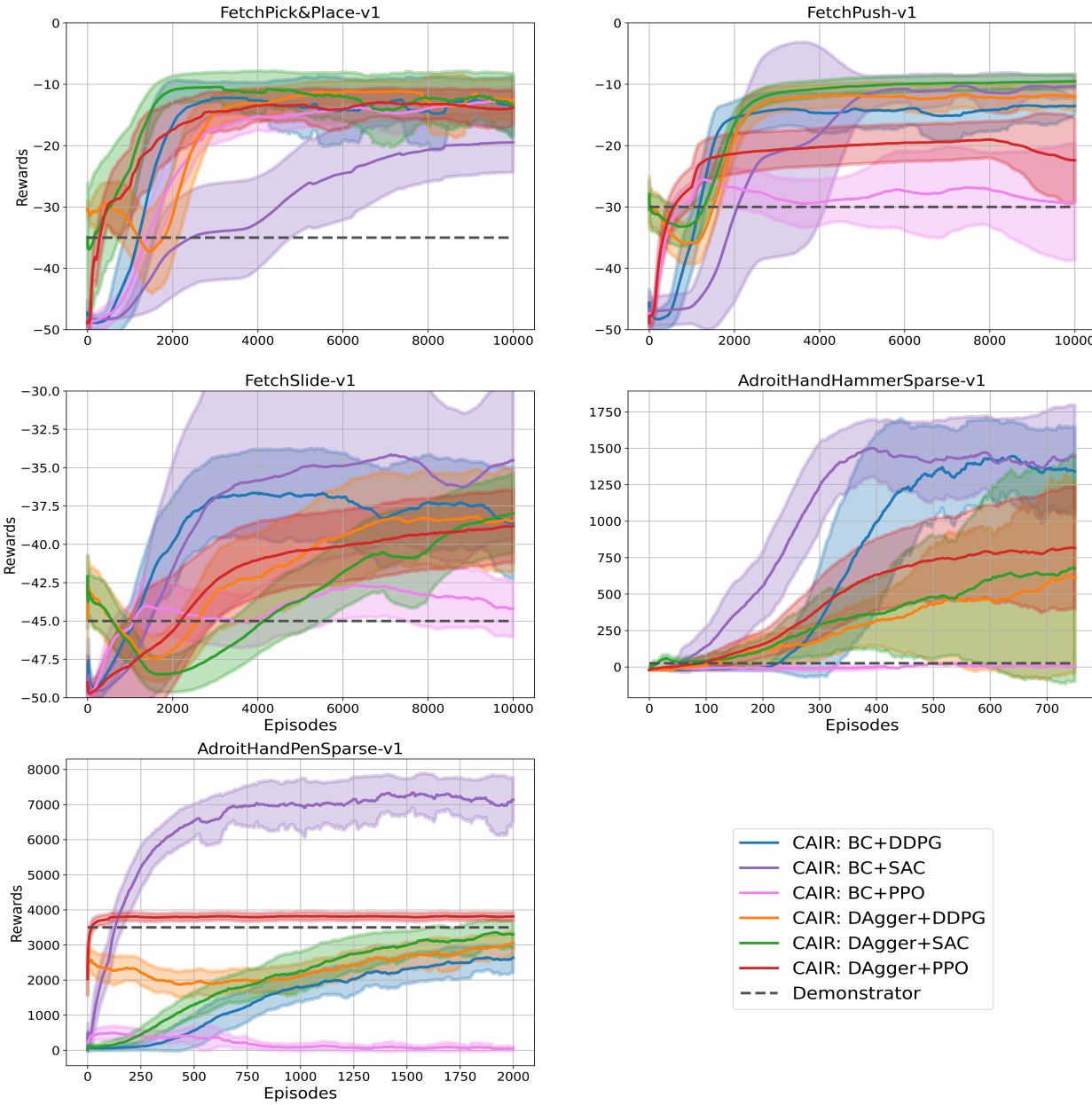

Figure 5: Learning curves for CAIR instantiated with different combinations of IL algorithms (BC, DAgger) and RL algorithms (SAC, DDPG, PPO). Positive learning trends are observed for all evaluated IL–RL pairings except BC+PPO on *AdroitHandHammerSparse-v1* and *AdroitHandPenSparse-v1*.

work architectures, and buffer configurations are taken directly from the authors' released implementation (Zhu et al., 2022).

### A.5.4 Jump-Start Reinforcement Learning (JSRL)

JSRL initializes training using a guide policy that steers the agent toward advantageous states at the start of each episode. The guide policy is gradually phased out by reducing the maximum guidance horizon. We use the formulation and hyperparameters from (Uchendu et al., 2022). The maximum guidance horizon is set to 40 for Fetch tasks (episode length 50) and 150 for AdroitHandHammerSparse (episode length 200).

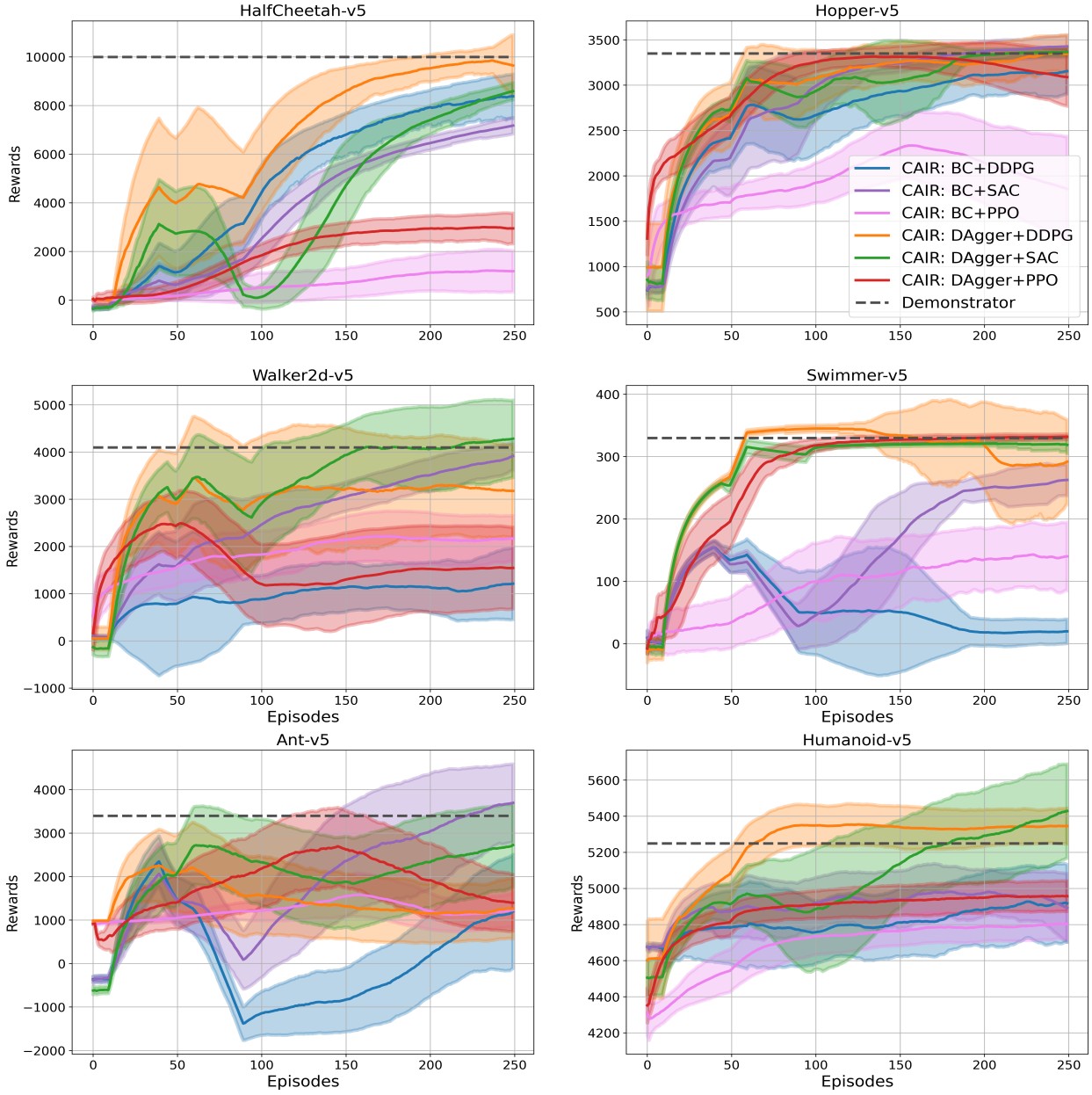

Figure 6: Learning curves for CAIR instantiated with different combinations of IL algorithms (BC, DAgger) and RL algorithms (SAC, DDPG, PPO) on D4RL locomotion tasks. Differences in learning performance are observed across IL–RL pairings, highlighting the impact of the underlying imitation and reinforcement learning algorithms.

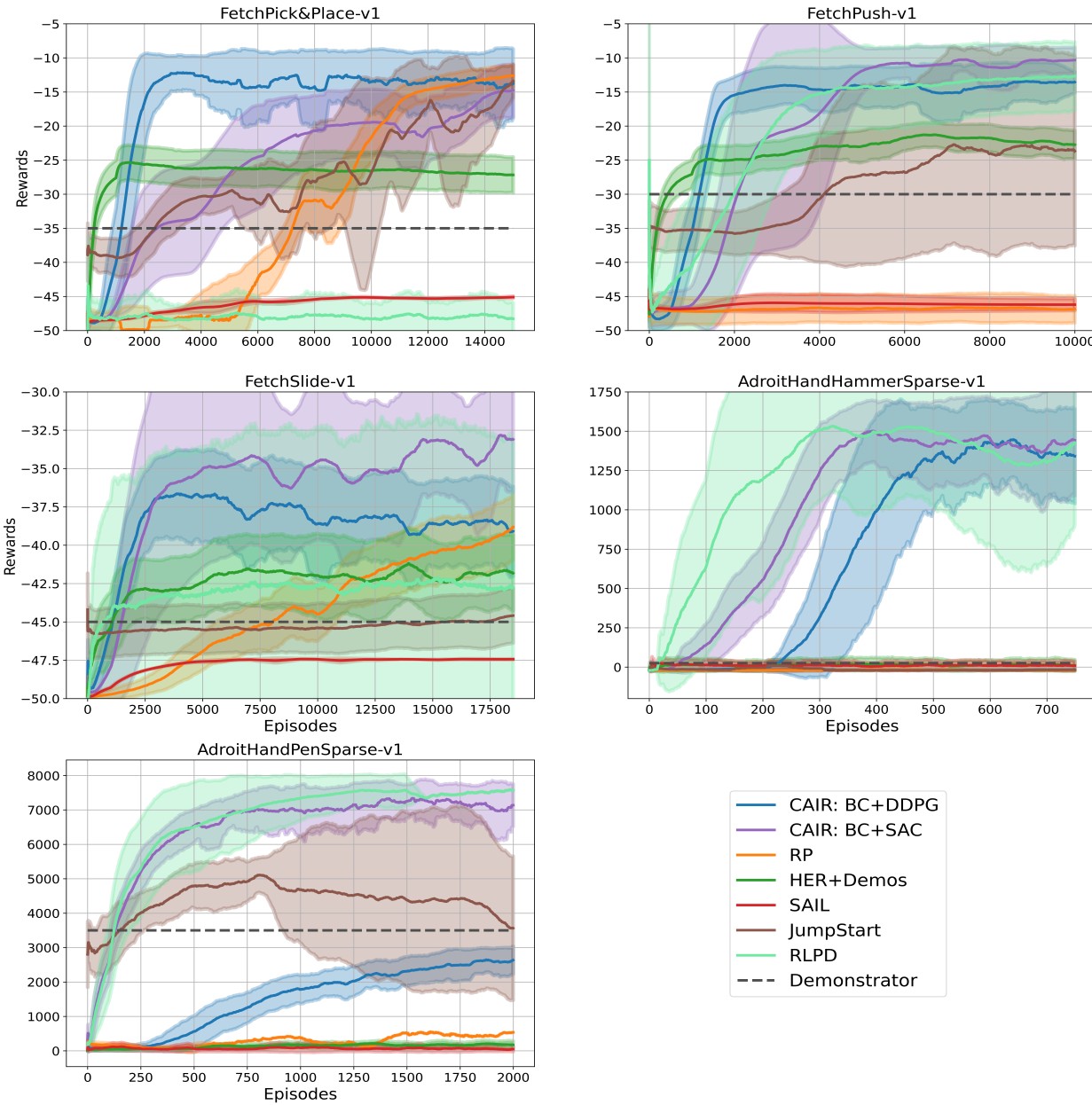

Figure 7: Learning trends for CAIR (BC+SAC and BC+DDPG) compared to hybrid-RL baselines on sparse-reward robotic manipulation tasks.

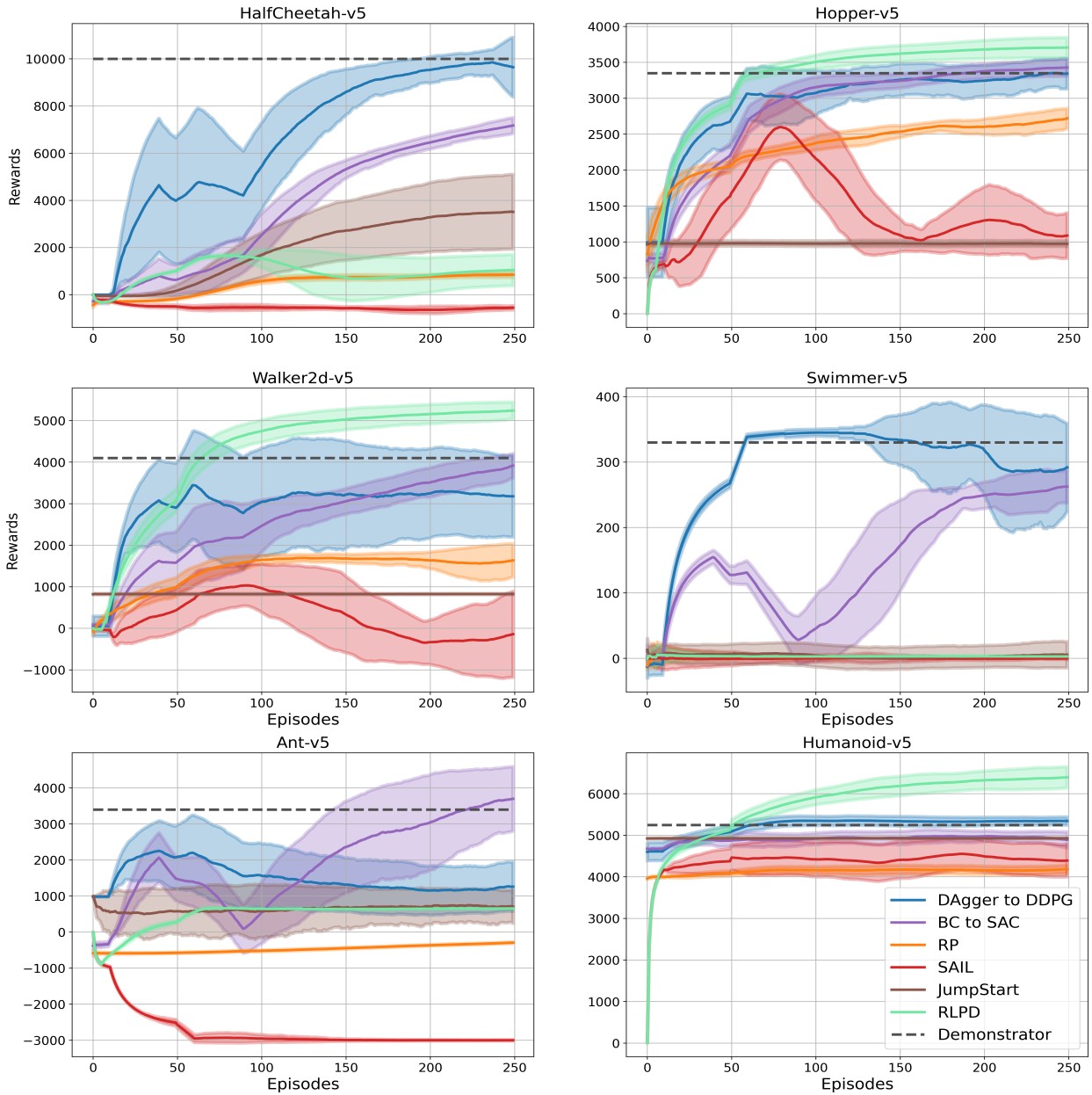

Figure 8: Learning trends for CAIR (BC+SAC and DAgger+DDPG) compared to hybrid-RL baselines on D4RL MuJoCo locomotion tasks.

Table 13: CAGrad performance ablation on Fetch manipulation tasks using matched no-CAGrad and CAGrad configurations. For each row, DAgger+DDPG, the annealing type, and the fixed rate $\delta$ are held constant. Values report final reward (mean $\pm$ standard deviation) and area under the learning curve over the first 8,000 episodes (AUC@8k). Within each matched pair, the preferred configuration is bolded. A gray result marked with † is significantly worse than its matched alternative according to a two-sided Welch's $t$-test ($p < 0.05$; $n = 5$ per configuration). When neither configuration fails the test, both are treated as comparable and the configuration with the higher AUC@8k is bolded. The final column reports the paired improvement in mean worst-case cosine, $\Delta c_{\mathrm{worst}} = c_{\mathrm{worst}}^{+\mathrm{CAGrad}} - c_{\mathrm{worst}}^{\mathrm{No\ CAGrad}}$, as mean $\pm$ sample standard deviation across the adaptive schedule and four fixed rates for each task.

| Task | Annealing | $\delta$ | No CAGrad | | | +CAGrad | | | $\Delta c_{\mathrm{worst}}$ ↑ |
|---|---|---|---|---|---|---|---|---|---|
| | | | Mean | $\pm$ SD | AUC@8k | Mean | $\pm$ SD | AUC@8k | |
| *FetchPickAndPlace* (8k episodes) | Adaptive | – | **-14.52** | **$\pm$ 2.01** | **-20.70** | -20.11 | $\pm$ 12.20 | -30.52 | |
| | Fixed | 0.01 | -25.26 | $\pm$ 5.50 | -35.40 | **-16.96** | **$\pm$ 13.13** | **-31.81** | |
| | Fixed | 0.05 | -34.20 | $\pm$ 7.87 | -42.36 | **-28.87** | **$\pm$ 27.18** | **-39.07** | +0.025 $\pm$ 0.113 |
| | Fixed | 0.10 | -42.34 | $\pm$ 6.38 | -47.54 | **-36.06** | **$\pm$ 17.45** | **-45.99** | |
| | Fixed | 0.15 | -40.41 | $\pm$ 7.69 | -48.15 | **-35.70** | **$\pm$ 17.15** | **-46.33** | |
| *FetchPush* (8k episodes) | Adaptive | – | †-17.66 | $\pm$ 2.79 | -21.68 | **-10.70** | **$\pm$ 2.24** | **-20.22** | |
| | Fixed | 0.01 | †-12.50 | $\pm$ 1.50 | -20.27 | **-9.49** | **$\pm$ 1.07** | **-16.17** | |
| | Fixed | 0.05 | †-13.52 | $\pm$ 2.78 | -21.50 | **-9.72** | **$\pm$ 1.22** | **-18.89** | +0.091 $\pm$ 0.017 |
| | Fixed | 0.10 | †-15.50 | $\pm$ 2.50 | -25.29 | **-9.91** | **$\pm$ 1.25** | **-21.81** | |
| | Fixed | 0.15 | **-12.50** | **$\pm$ 2.37** | **-24.88** | †-46.42 | $\pm$ 2.19 | -46.61 | |
| *FetchSlide* (8k episodes) | Adaptive | – | †-43.17 | $\pm$ 2.85 | -45.21 | **-36.54** | **$\pm$ 5.34** | **-37.73** | |
| | Fixed | 0.01 | -43.50 | $\pm$ 3.89 | -46.23 | -38.81 | $\pm$ 5.38 | -43.79 | |
| | Fixed | 0.05 | †-39.42 | $\pm$ 3.46 | -42.63 | **-32.94** | **$\pm$ 1.85** | **-35.63** | +0.100 $\pm$ 0.016 |
| | Fixed | 0.10 | **-42.50** | **$\pm$ 2.50** | **-45.53** | †-49.35 | $\pm$ 0.00 | -48.18 | |
| | Fixed | 0.15 | **-41.50** | **$\pm$ 2.80** | **-44.50** | -44.87 | $\pm$ 3.99 | -48.49 | |
| | | | | | | | | *Across all 15 matched configurations* | +0.072 $\pm$ 0.070 |

*Visual key:* **bold black** = preferred within the matched pair; regular black = statistically comparable but not selected by AUC; † gray = significantly worse under the Welch's $t$-test. For +CAGrad across all 15 comparisons: **best in 11 (73.3%)**, comparable in 2 (13.3%), and significantly worse in 2 (13.3%).

Table 14: Adaptive-annealing ablation on D4RL locomotion tasks with DAgger+DDPG and CAGrad disabled. Values report final reward (mean ± standard deviation) and area under the learning curve over the first 250 episodes (AUC@250). Methods marked with † are significantly worse than the highest-mean schedule for the same task according to a two-sided Welch's $t$-test ($p < 0.05$; $n = 10$ per configuration). Among the remaining schedules, the method with the highest AUC@250 is bolded.

| Task | Annealing | $\delta$ | Mean | ± SD | AUC@250 |
|------|-----------|----------|------|------|---------|
| *HalfCheetah* (250k steps) | Adaptive | – | 10,980 | ± 644 | 4,728 |
| | **Fixed** | **0.01** | **10,590** | **± 78** | **9,262** |
| | Fixed | 0.05 | 10,548 | ± 106 | 9,211 |
| | Fixed | 0.10 | †4,235 | ± 432 | 7,398 |
| | Fixed | 0.15 | †5,324 | ± 990 | 6,094 |
| *Hopper* (250k steps) | **Adaptive** | – | **3,594** | **± 150** | **2,921** |
| | Fixed | 0.01 | †3,425 | ± 7 | 3,171 |
| | Fixed | 0.05 | †3,429 | ± 9 | 3,148 |
| | Fixed | 0.10 | †2,952 | ± 168 | 3,018 |
| | Fixed | 0.15 | †2,332 | ± 1,191 | 2,763 |
| *Walker2d* (250k steps) | Adaptive | – | †3,563 | ± 432 | 1,976 |
| | Fixed | 0.01 | 4,899 | ± 144 | 4,310 |
| | **Fixed** | **0.05** | **4,834** | **± 99** | **4,329** |
| | Fixed | 0.10 | †2,756 | ± 229 | 3,716 |
| | Fixed | 0.15 | †2,347 | ± 664 | 3,047 |
| *Swimmer* (250k steps) | **Adaptive** | – | **345.3** | **± 2.5** | **275.1** |
| | Fixed | 0.01 | †335.7 | ± 1.5 | 300.1 |
| | Fixed | 0.05 | †334.2 | ± 1.2 | 298.0 |
| | Fixed | 0.10 | †69.3 | ± 14.2 | 229.7 |
| | Fixed | 0.15 | †19.1 | ± 21.8 | 158.9 |
| *Ant* (250k steps) | Adaptive | – | †1,394 | ± 801 | -444 |
| | **Fixed** | **0.01** | **3,489** | **± 333** | **3,053** |
| | Fixed | 0.05 | 3,086 | ± 906 | 2,968 |
| | Fixed | 0.10 | †-464 | ± 443 | 1,919 |
| | Fixed | 0.15 | †-1,062 | ± 511 | 969 |
| *Humanoid* (250k steps) | Adaptive | – | 5,128 | ± 119 | 4,822 |
| | Fixed | 0.01 | †5,042 | ± 11 | 5,017 |
| | **Fixed** | **0.05** | **5,057** | **± 11** | **5,022** |
| | Fixed | 0.10 | †4,795 | ± 148 | 4,953 |
| | Fixed | 0.15 | †4,759 | ± 218 | 4,882 |

Table 15: CAGrad performance ablation on D4RL locomotion tasks using matched no-CAGrad and CAGrad configurations. For each row, DAgger+DDPG, the annealing type, and the fixed rate $\delta$ are held constant. Values report final reward (mean ± standard deviation) and area under the learning curve over the first 250 episodes (AUC@250). Within each matched pair, the preferred configuration is bolded. A gray result marked with † is significantly worse than its matched alternative according to a two-sided Welch's $t$-test ($p < 0.05$; $n = 10$ per configuration). When neither configuration fails the test, both are treated as comparable and the configuration with the higher AUC@250 is bolded. The final column reports the paired improvement in mean worst-case cosine, $\Delta c_{\text{worst}} = c_{\text{worst}}^{+\text{CAGrad}} - c_{\text{worst}}^{\text{No CAGrad}}$, as mean ± sample standard deviation across the adaptive schedule and four fixed rates for each task. Positive values indicate improved alignment with the less-aligned objective gradient.

| Task | Annealing | $\delta$ | No CAGrad | | | +CAGrad | | | $\Delta c_{\text{worst}}$ ↑ |
| | | | Mean | ± SD | AUC@250 | Mean | ± SD | AUC@250 | |
|---|---|---|---|---|---|---|---|---|---|
| *HalfCheetah* (250k steps) | Adaptive | – | 10,980 | ± 644 | 4,728 | **10,994** | **± 216** | **7,082** | |
| | Fixed | 0.01 | **10,590** | **± 78** | **9,262** | †9,578 | ± 521 | 1,220 | |
| | Fixed | 0.05 | †10,548 | ± 106 | 9,211 | **10,930** | **± 383** | **3,579** | +0.088 ± 0.099 |
| | Fixed | 0.10 | †4,235 | ± 432 | 7,398 | **10,787** | **± 650** | **3,961** | |
| | Fixed | 0.15 | †5,324 | ± 990 | 6,094 | **10,115** | **± 2,127** | **4,213** | |
| *Hopper* (250k steps) | Adaptive | – | **3,594** | **± 150** | **2,921** | †3,424 | ± 156 | 2,974 | |
| | Fixed | 0.01 | **3,425** | **± 7** | **3,171** | 3,430 | ± 181 | 2,979 | |
| | Fixed | 0.05 | **3,429** | **± 9** | **3,148** | 3,225 | ± 573 | 3,089 | +0.325 ± 0.060 |
| | Fixed | 0.10 | †2,952 | ± 168 | 3,018 | **3,520** | **± 172** | **2,818** | |
| | Fixed | 0.15 | †2,332 | ± 1,191 | 2,763 | **3,721** | **± 185** | **2,871** | |
| *Walker2d* (250k steps) | Adaptive | – | 3,563 | ± 432 | 1,976 | **3,946** | **± 869** | **3,011** | |
| | Fixed | 0.01 | **4,899** | **± 144** | **4,310** | †2,317 | ± 302 | 4,308 | |
| | Fixed | 0.05 | **4,834** | **± 99** | **4,329** | †2,531 | ± 854 | 1,675 | +0.156 ± 0.068 |
| | Fixed | 0.10 | †2,756 | ± 229 | 3,716 | **3,541** | **± 630** | **1,632** | |
| | Fixed | 0.15 | †2,347 | ± 664 | 3,047 | **3,818** | **± 506** | **1,690** | |
| *Swimmer* (250k steps) | Adaptive | – | 345.3 | ± 2.5 | 275.1 | **334.8** | **± 19.1** | **290.6** | |
| | Fixed | 0.01 | **335.7** | **± 1.5** | **300.1** | †13.5 | ± 18.2 | 301.1 | |
| | Fixed | 0.05 | **334.2** | **± 1.2** | **298.0** | †32.7 | ± 26.4 | 135.4 | +0.314 ± 0.091 |
| | Fixed | 0.10 | **69.3** | **± 14.2** | **229.7** | †42.7 | ± 12.4 | 98.4 | |
| | Fixed | 0.15 | **19.1** | **± 21.8** | **158.9** | 124.9 | ± 196.8 | 103.8 | |
| *Ant* (250k steps) | Adaptive | – | 1,394 | ± 801 | -444 | **1,014** | **± 350** | **1,425** | |
| | Fixed | 0.01 | **3,489** | **± 333** | **3,053** | †-1,111 | ± 381 | 1,580 | |
| | Fixed | 0.05 | **3,086** | **± 906** | **2,968** | †736 | ± 1,134 | -1,143 | +0.291 ± 0.041 |
| | Fixed | 0.10 | †-464 | ± 443 | 1,919 | **1,315** | **± 346** | **-983** | |
| | Fixed | 0.15 | †-1,062 | ± 511 | 969 | **1,275** | **± 999** | **-927** | |
| *Humanoid* (250k steps) | Adaptive | – | †5,128 | ± 119 | 4,822 | **5,526** | **± 175** | **5,270** | |
| | Fixed | 0.01 | **5,042** | **± 11** | **5,017** | †4,711 | ± 165 | 5,005 | |
| | Fixed | 0.05 | **5,057** | **± 11** | **5,022** | 4,981 | ± 259 | 4,913 | +0.035 ± 0.041 |
| | Fixed | 0.10 | **4,795** | **± 148** | **4,953** | 4,905 | ± 514 | 4,802 | |
| | Fixed | 0.15 | †4,759 | ± 218 | 4,882 | **5,098** | **± 370** | **4,762** | |
| | | | | | | | *Across all 30 matched configurations* | | +0.202 ± 0.133 |

*Visual key:* **bold black** = preferred within the matched pair; regular black = statistically comparable but not selected by AUC; † gray = significantly worse under the Welch's $t$-test. For +CAGrad across all 30 comparisons: **best in 15 (50.0%)**, comparable in 5 (16.7%), and significantly worse in 10 (33.3%).

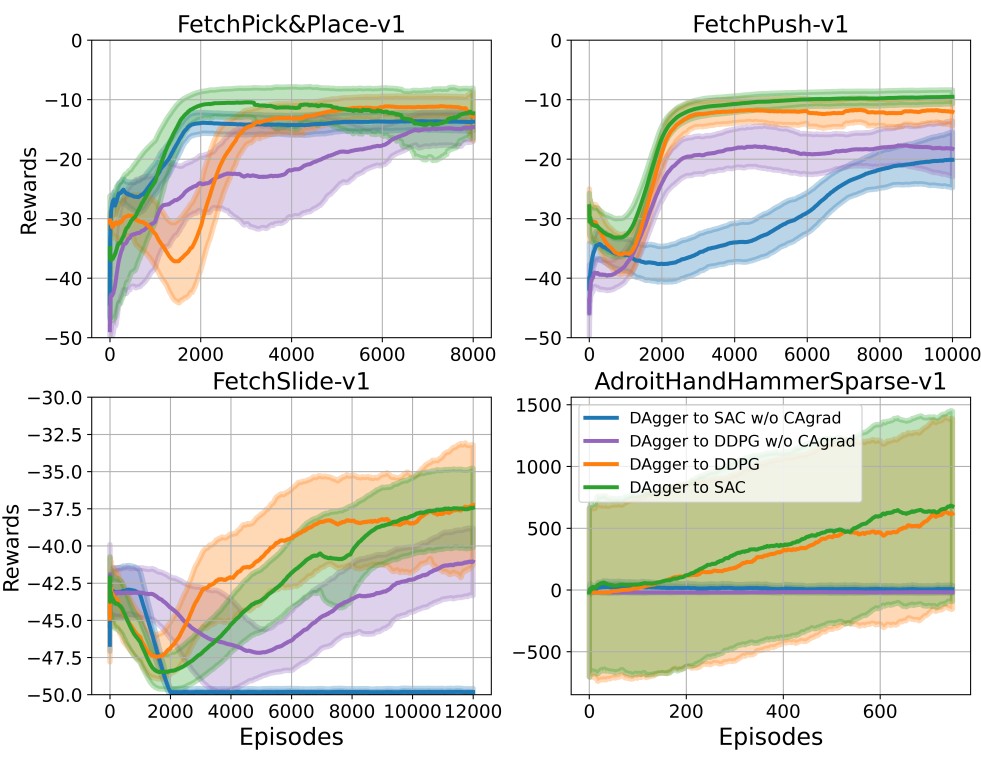

Figure 9: Ablation study evaluating the effect of CAGrad on learning performance across tasks. Improvements are observed in sample efficiency and asymptotic performance. (Fetch: 1 ep = 1k steps, Adroit: 1 ep = 4k steps).

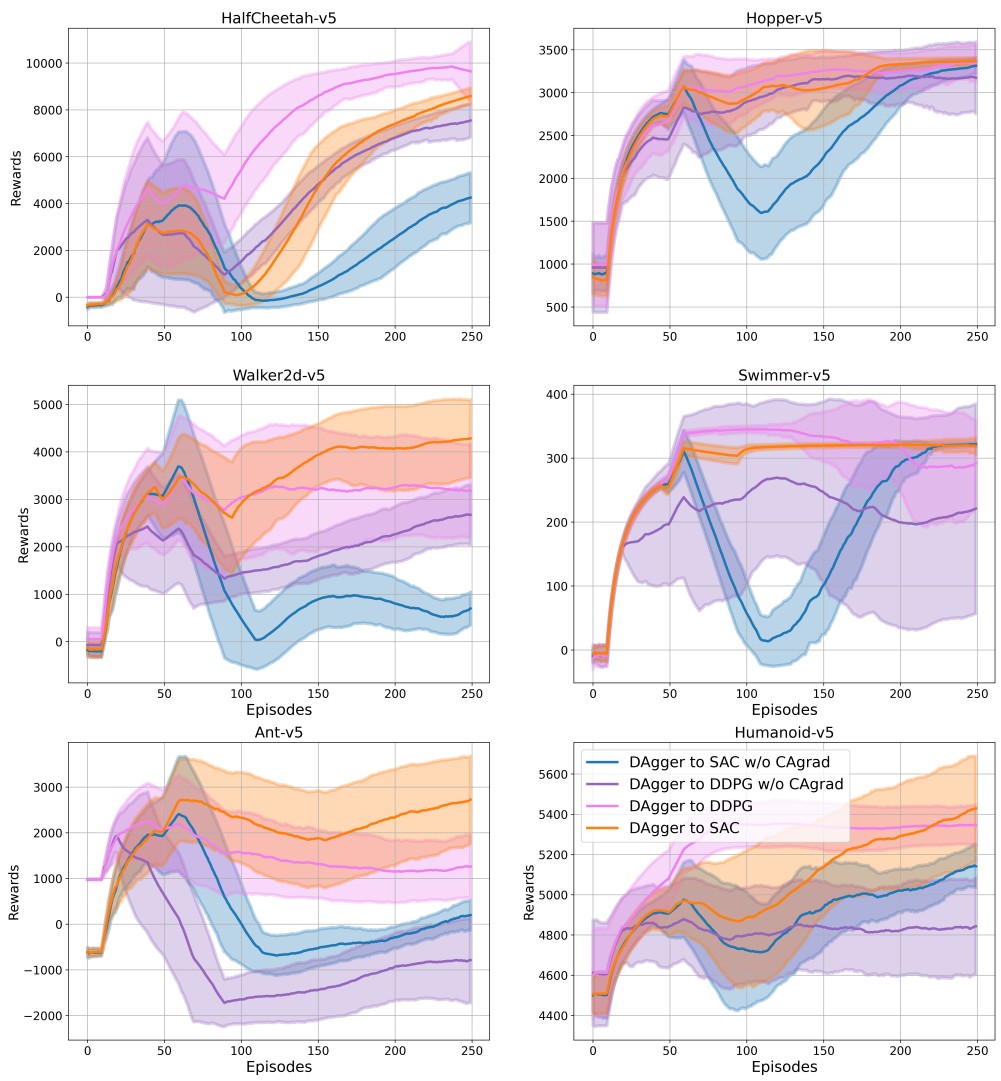

Figure 10: Ablation study comparing learning performance with and without CAGrad across D4RL locomotion tasks. Adding CAGrad notably improves the learning curves, reducing pronounced mid-training return drops and producing higher final returns. (Locomotion: 1 episode = 1k steps).

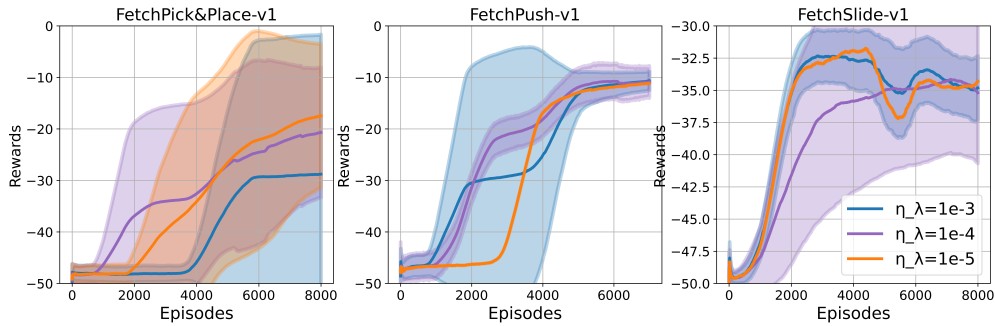

Figure 11: Sensitivity analysis: Effect of $\lambda$-optimizer learning rate $(\eta_\lambda)$ on Fetch tasks.

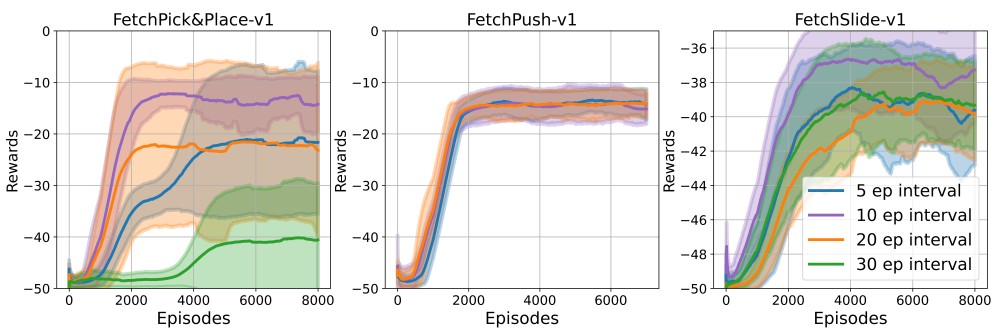

Figure 12: Sensitivity analysis: Effect of $\lambda$ update interval on Fetch tasks.

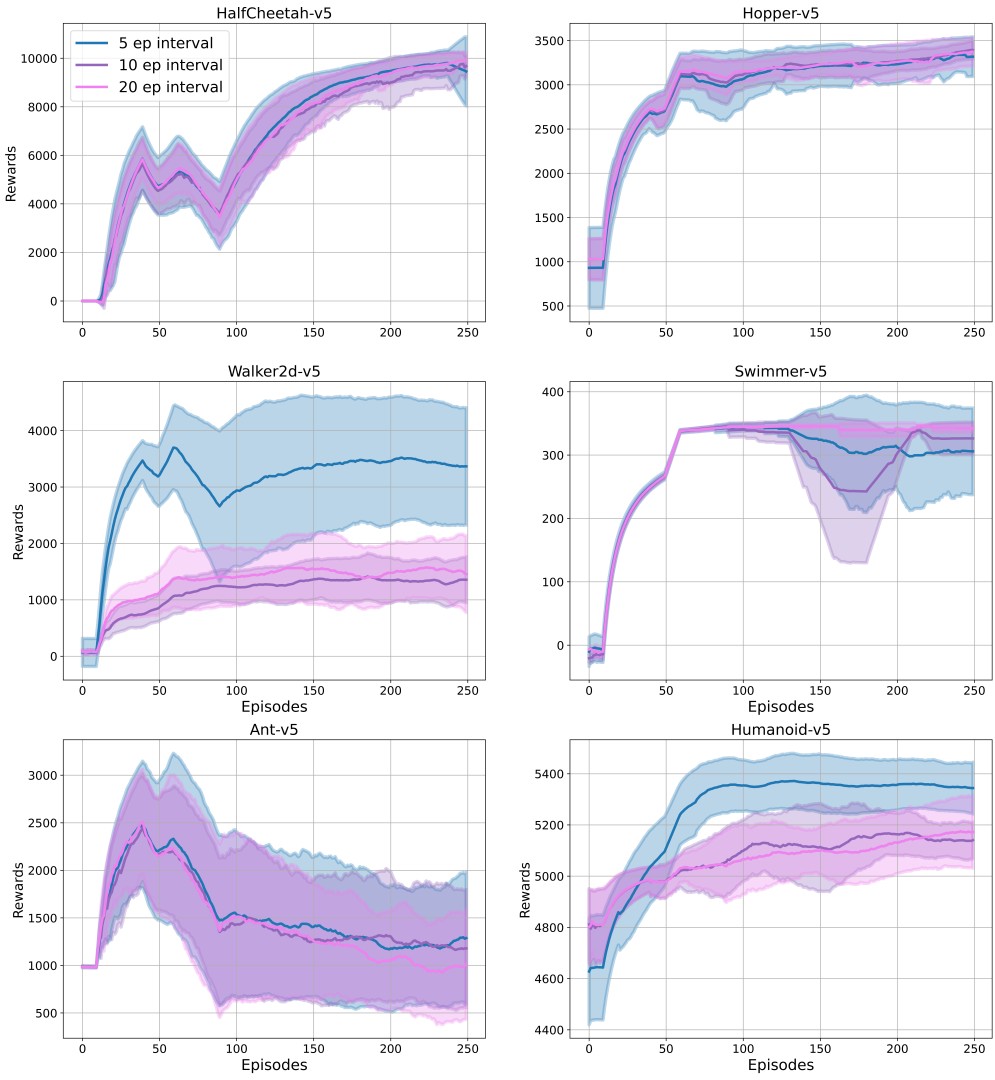

Figure 13: Sensitivity analysis: Effect of $\lambda$ update interval on D4RL MuJoCo locomotion tasks.

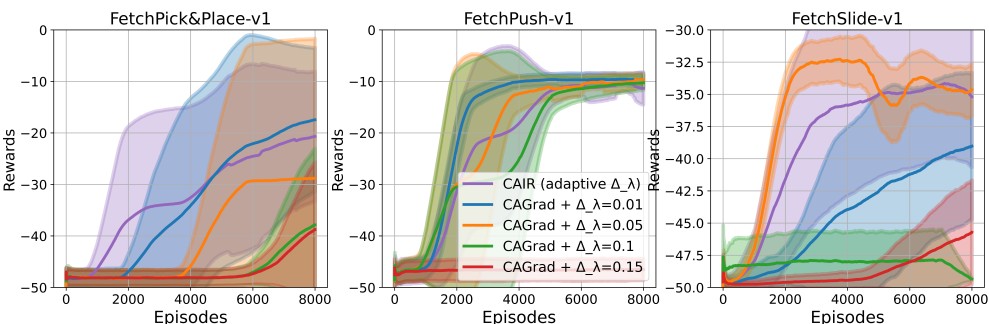

Figure 14: Ablation study comparing adaptive annealing in CAIR with fixed schedules $\lambda \leftarrow \lambda - \Delta$ across the Fetch tasks, with CAGrad enabled for all variants. Adaptive annealing is favored or statistically comparable to the strongest tested fixed schedule in all three tasks while avoiding task-specific selection of $\Delta$; larger fixed rates also produce substantial degradation in several settings.

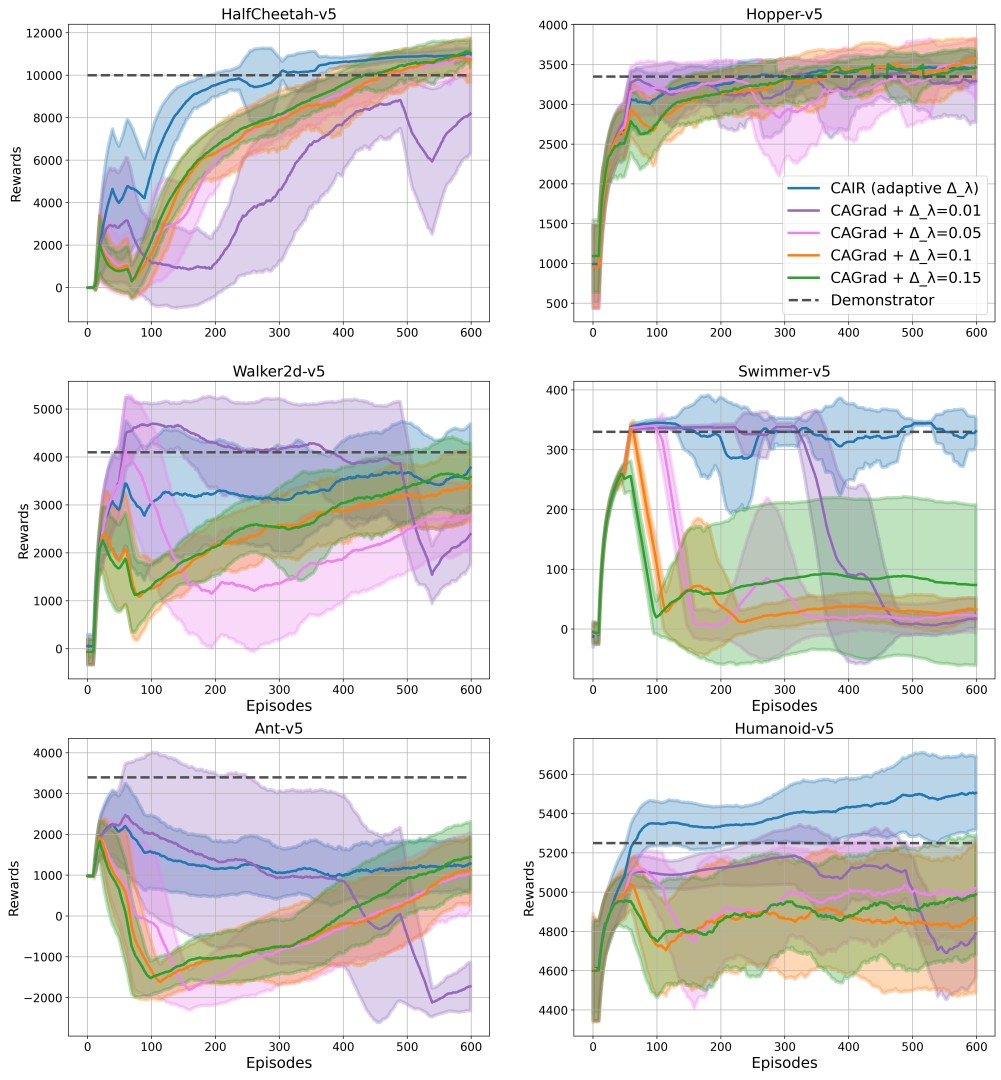

Figure 15: Ablation study comparing adaptive annealing (CAIR) with fixed schedule annealing $\lambda \leftarrow \lambda - \Delta$ across D4RL locomotion tasks. Adaptive annealing achieves improved sample efficiency and smoother reward progression, whereas fixed schedules often exhibit performance drops near the end of annealing before recovering.

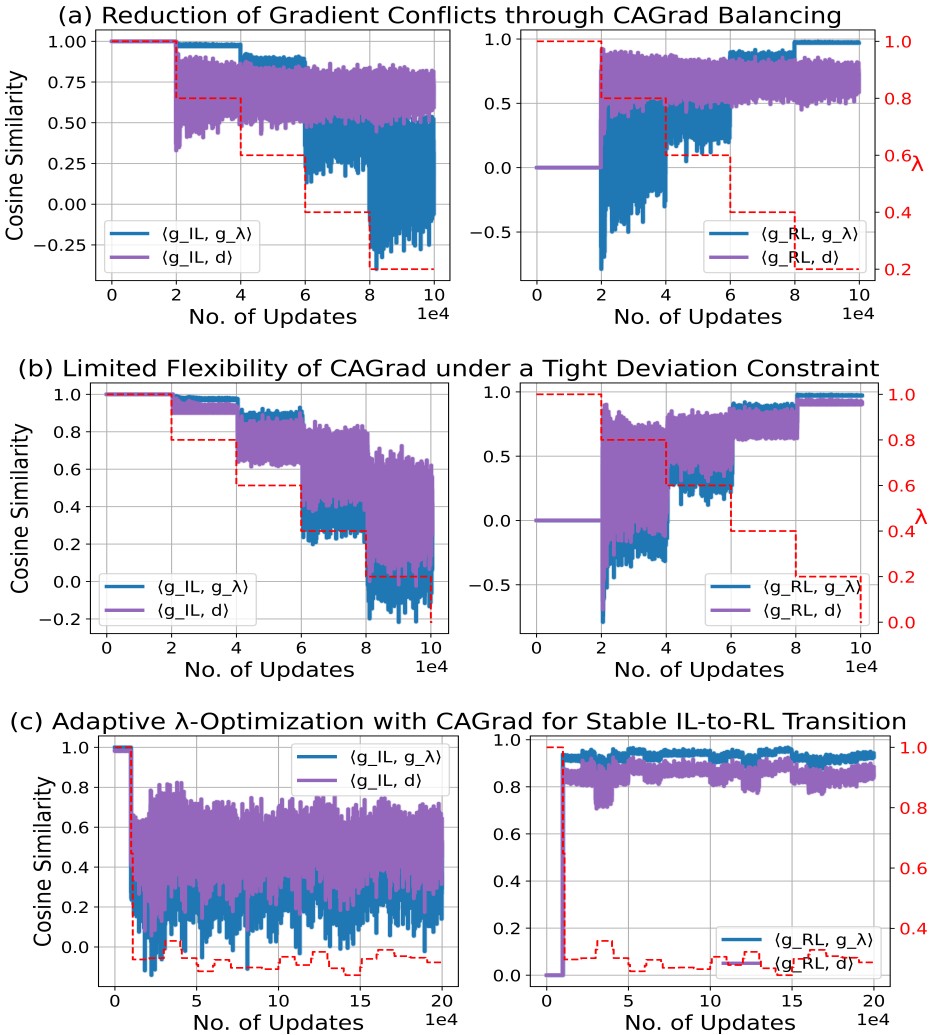

Figure 16: Cosine similarity analysis on *FetchPush-v1* during the IL-to-RL transition. Each of the six plots reports the alignment between the combined update direction and the task-specific (IL, RL) gradients: cosine similarity with $g_{\mathrm{IL}}$ (left column) and with $g_{\mathrm{RL}}$ (right column). **(a)** CAGrad resolves gradient conflict during fixed schedule annealing, but requires tuning. **(b)** CAGrad with a tight constraint fails to resolve gradient conflict under fixed annealing. **(c)** CAIR (Adaptive Annealing + CAGrad) maintains positive alignment, even under tight constraints. (x-axis: gradient updates; y-axis: cosine similarity averaged over 10 updates).

